# Multi-trait analysis of rare-variant association summary statistics using MTAR

Lan Luo[1,7], Judong Shen [2,7], Hong Zhang [2], Aparna Chhibber[3], Devan V. Mehrotra[4] & Zheng-Zheng Tang [5,6✉]

Integrating association evidence across multiple traits can improve the power of gene discovery and reveal pleiotropy. Most multi-trait analysis methods focus on individual common variants in genome-wide association studies. Here, we introduce multi-trait analysis of rare-variant associations (MTAR), a framework for joint analysis of association summary statistics between multiple rare variants and different traits. MTAR achieves substantial power gain by leveraging the genome-wide genetic correlation measure to inform the degree of gene-level effect heterogeneity across traits. We apply MTAR to rare-variant summary statistics for three lipid traits in the Global Lipids Genetics Consortium. 99 genome-wide significant genes were identified in the single-trait-based tests, and MTAR increases this to 139. Among the 11 novel lipid-associated genes discovered by MTAR, 7 are replicated in an independent UK Biobank GWAS analysis. Our study demonstrates that MTAR is substantially more powerful than single-trait-based tests and highlights the value of MTAR for novel gene discovery.

[1] Department of Statistics, University of Wisconsin-Madison, Madison, Wisconsin 53706, USA. [2] Biostatistics and Research Decision Sciences, Merck & Co., Inc., Rahway, New Jersey 07065, USA. [3] Genetics and Pharmacogenomics, Merck & Co., Inc., West Point, Pennsylvania 19446, USA. [4] Biostatistics and Research Decision Sciences, Merck & Co., Inc., North Wales, Pennsylvania 19454, USA. [5] Department of Biostatistics and Medical Informatics, University of Wisconsin-Madison, Madison, Wisconsin 53715, USA. [6] Wisconsin Institute for Discovery, Madison, Wisconsin 53715, USA. [7]These authors contributed equally: Lan Luo, Judong Shen. ✉email: tang@biostat.wisc.edu

Rich genome-wide association study (GWAS) findings have suggested the sharing of genetic risk variants among multiple complex traits[1,2]. Multi-trait analyses that combine association evidence across traits can boost statistical power over single-trait analyses in detecting risk variants, especially for those traits that have weak associations with the variants. Many multi-trait methods are designed for testing the single-variant association[3–8]. However, the statistical power of single-variant tests is low for rare-variant association studies (RVAS)[9]. In light of this limitation, gene-based tests have been developed for RVAS to aggregate mutation information across several variant sites within a gene to enrich association signals and reduce the penalty resulting from multiple testing[9]. Although several methods are available for multi-trait multi-variant tests, most of them require individual-level genotype and phenotype data[10–15] or are designed for common variants[16–19] (Supplementary Table 1). The gene-based tests for RVAS have not been fully exploited in the multi-trait analysis.

The genetic architecture of complex traits is unknown in advance and is likely to vary from one gene to another across the genome and from one trait to another. Therefore, the main challenge of multi-trait multi-variant analyses is to flexibly accommodate a variety of genetic effect patterns among traits and variants such that the test is robust and has high power. The effect structures among rare variants within a gene have been well-studied when numerous gene-based tests were developed. The sequence kernel association test (SKAT)[20] and burden tests[21–24] are the most widely used gene-based tests for RVAS and represent two main patterns of genetic effects across rare variants. Burden tests assume effects across variants are largely homogeneous and SKAT assumes they are heterogeneous. SKAT-O[25] is a test that achieves robustness by combining tests with various degrees of effect heterogeneity, including the SKAT and burden tests as special cases. Specifically, SKAT-O assumes rare-variant effects are random variables with a uniform (exchangeable) correlation and different levels of heterogeneity can be considered by changing the correlation coefficient.

The effects on multiple traits may also exhibit homogeneous and heterogeneous patterns. However, the degree of genetic effect similarity/heterogeneity are likely to vary from one trait pair to another. As an example, for the pair of traits that are biologically related (e.g., triglycerides (TG) and high-density lipoprotein cholesterol (HDL)), we expect they share more causal variants and have a higher level of genetic similarity than the pair of traits less relevant (e.g., TG and bipolar)[26]. Hence, it is not adequate to use a uniform correlation coefficient to model the degree of similarity for all trait pairs. Many recent studies have investigated the genetic overlap for many pairs of complex traits and diseases and estimated genetic correlation as a global measure of genetic similarity for trait pairs[26–28]. Although a genetic correlation is calculated using common variants across the genome and RV association tests are performed on the gene level, the idea of utilizing genetic correlation to guide the specification of gene-level effect heterogeneity across traits is intriguing and has not been considered in existing multi-trait methods.

Here we develop multi-trait analysis of rare-variant association (MTAR), a framework for the multi-trait analysis of RVAS. MTAR is built upon a random-effects meta-analysis model that uses different correlation structures of the genetic effects to represent a wide spectrum of association patterns across traits and variants. To model genetic effects across variants, MTAR employs the same strategy as SKAT-O. To model the rare-variant effect heterogeneity on multiple traits, MTAR leverages the genetic correlation. Specifically, we propose two correlation structures on the among-trait genetic effects. The first structure allows the between-trait effect similarity to change from the value

of the genetic correlation to completely heterogeneous as an extreme and we term the resulting multi-trait association test iMTAR. The second structure allows the between-trait effect similarity to change from the value of the genetic correlation to homogeneous as an extreme and we term the resulting test cMTAR. Besides the aforementioned association patterns across traits, we also consider the scenario in which only a small number of traits are associated with the set of rare variants. This association pattern naturally occurs for the genes that have very specific biological functions and do not affect many traits. To accommodate this pattern, we construct another test, cctP, which uses the Cauchy method[29,30] to combine single-trait RVAS P-values. To achieve robustness and improve overall power, we combine the P-values of iMTAR, cMTAR, and cctP, and refer to this omnibus test as MTAR-O. To demonstrate the usefulness of MTAR empirically, we analyze summary statistics from the Global Lipids Genetics Consortium (GLGC) on low-density lipoprotein cholesterol (LDL), HDL, and TG. MTAR discovers more lipid-associated genes than single-trait-based analyses and many novel association signals are replicated in an independent UK Biobank data. Moreover, our simulation results show that MTAR methods have well-preserved type I error rate and greater power over single-trait-based methods across a wide range of effect patterns across traits and variants. Finally, we compare MTAR with two existing multi-trait methods that outperform other competing methods. We find that MTAR is more powerful in almost all simulation settings and discovers more genes in the application to the GLGC data.

## Results

**MTAR overview.** Suppose that we are interested in the effects of $m$ variants in a gene on $K$ traits. For $k = 1, …, K$, we let $\boldsymbol{\beta}_k = (\beta_{k1}, …, \beta_{km})^T$ denote the effects of the $m$ genetic variables on trait $k$. To perform MTAR tests, we first obtain the vector of variant-level score statistics for testing $\boldsymbol{\beta}_k = \mathbf{0}$ denoted by $\mathbf{U}_k = (U_{k1}, …, U_{km})^T$ and the covariance estimate for $\mathbf{U}_k$ denoted by $\mathbf{V}_k$. The $\mathbf{U}_k$ and $\mathbf{V}_k$ can be easily constructed using the information routinely shared in public domains (Methods). We let $\widehat{\boldsymbol{\beta}}_k = \mathbf{V}_k^{-1}\mathbf{U}_k$ and write $\widehat{\boldsymbol{\beta}} = (\widehat{\boldsymbol{\beta}}_1^T, …, \widehat{\boldsymbol{\beta}}_K^T)^T$. Given the true genetic effects $\boldsymbol{\beta} = (\boldsymbol{\beta}_1^T, …, \boldsymbol{\beta}_K^T)^T$, the $\widehat{\boldsymbol{\beta}}$ approximately follows normal distribution with mean $\boldsymbol{\beta}$ and covariance $\boldsymbol{\Sigma}$[31,32], where $\boldsymbol{\Sigma} = \text{Blockdiag}\{\mathbf{V}_1^{-1}, …, \mathbf{V}_K^{-1}\}$ if traits are measured on studies without overlapping samples. If all the traits are from one study or multiple studies with overlapping subjects, the off-diagonal blocks in $\boldsymbol{\Sigma}$ are not zeros. For any given traits $k$ and $k'$ with sample overlap, the formula for estimating the covariance between $\widehat{\boldsymbol{\beta}}_k$ and $\widehat{\boldsymbol{\beta}}_{k'}$ is provided in Eq. (3).

We are interested in testing the null hypothesis that the $m$ variants are not associated with any of the $K$ traits: $H_0 : \boldsymbol{\beta}_1 = \boldsymbol{\beta}_2 = \cdots = \boldsymbol{\beta}_K = \mathbf{0}$. Multivariate test for this hypothesis has a large degrees of freedom and low statistical power. In MTAR, we further assume that the genetic effects $\boldsymbol{\beta}$ are zero-mean random effects with covariance matrix $\sigma\mathbf{B}$, where $\sigma$ is an unknown scalar and $\mathbf{B}$ is a pre-specified matrix dictating the covariances of genetic effects among traits and variants. Under this random-effects model, the equivalent null hypothesis is $H_0 : \sigma = 0$ and we test this hypothesis using a variance-component score test (Eq. (5)). The test will have the optimal power if the specification of $\mathbf{B}$ reflects the true covariance structure of the effects. The true structure of $\mathbf{B}$ is unknown a priori. To separately model the genetic structures among trait and among variants, we propose to formulate $\mathbf{B} = \mathbf{B}_2 \otimes \mathbf{B}_1$, where $\otimes$ is the Kronecker product of among-variant effect covariance $\mathbf{B}_1$ and among-trait effect covariance $\mathbf{B}_2$. For $\mathbf{B}_1$, we assume the exchangeable correlation

structure with a uniform correlation coefficient denoted by $\rho_1$ (Methods). By specifying different values of $\rho_1$, this structure allows various degrees of among-variant effect heterogeneity. As the two extremes, the effects across variants are homogeneous when $\rho_1 = 1$; the effects are completely heterogeneous and vary independently when $\rho_1 = 0$.

For the between-trait effect covariance, we set $\mathbf{B}_2 = \mathbf{W}_2 \mathbf{\Omega}_2 \mathbf{W}_2$, where $\mathbf{W}_2$ is a diagonal matrix with each diagonal element being a trait-specific weight and $\mathbf{\Omega}_2$ is a between-trait effect correlation matrix. By setting the diagonal elements in $\mathbf{W}_2$ to 0 or 1, we can choose to focus on any subset of the traits and consider any degree of association sparsity across traits (e.g., set only one element as 1 for single-trait analysis or all the elements as 1 for all-trait analysis). It is not sensible to assume the exchangeable correlation structure for $\mathbf{B}_2$, because some pairs of traits are more similar in the rare-variant effects than other pairs (e.g., two diseases that were caused by the same set of rare mutations would have a large correlation in their rare-variant genetic effects). Here we propose to leverage the genetic correlation[27] to inform the similarity of rare-variant effects among traits. Genetic correlation is a single number measure that quantifies the overall genetic similarity between a pair of traits. Recent advancement of methods enables us to conveniently estimate genetic correlation based on GWAS summary statistics[27,28] and there are web portals to query genetic correlations among many complex traits[26]. We hypothesize that the genetic correlation is also informative to measure the similarity/heterogeneity of the gene-level rare-variant effects among traits for most genes in the genome. Specifically, let $C_{kk'}$ denote the genetic correlation between traits $k$ and $k'$. We propose two types of correlation structures for $\mathbf{\Omega}_2$. In both

structures, we specify a parameter $\rho_2$ ($0 \le \rho_2 \le 1$) to control the contribution of genetic correlation $C_{kk'}$ to the degree of effect heterogeneity between traits $k$ and $k'$. The iMTAR structure assumes the correlation coefficient is $\rho_2 C_{kk'}$. Under this structure, the rare-variant effects across traits are heterogeneous and the degree of heterogeneity can change from $C_{kk'}$ (when $\rho_2 = 1$) to completely heterogeneous (strongest level of heterogeneity as effects across traits can vary independently when $\rho_2 = 0$). The cMTAR structure assumes the correlation coefficient is $\rho_2 C_{kk'} + (1 - \rho_2)$. Under this structure, the degree of heterogeneity can change from $C_{kk'}$ (when $\rho_2 = 1$) to homogeneous (no heterogeneity when $\rho_2 = 0$).

As the optimal values of $\rho_1$ and $\rho_2$ are unknown, we propose to search a grid of different values of $\rho_1$ and $\rho_2$ and use the Cauchy method[29,30] to combine multiple $P$-values (Methods). The resulting tests are named after the two aforementioned iMTAR and cMTAR structures. The Cauchy method is a fast and powerful approach to combine multiple correlated $P$-values without the need for estimating and accounting for their correlation. To accommodate the situation where the gene is associated with a small number of traits, we develop a test called cctP that uses the Cauchy method to combine single-trait $P$-values from SKAT and burden tests. As we demonstrate in the GLGC data analysis and simulation studies, the cMTAR, iMTAR, and cctP cover different effect patterns among traits. To achieve further robustness, we use the Cauchy method to combine $P$-values of the three complementary tests and term this omnibus test as MTAR-O. The summary of the proposed iMTAR, cMTAR, cctP, and MTAR-O methods are presented in Fig. 1. The calculations of the test statistics and $P$-values are described in Methods.

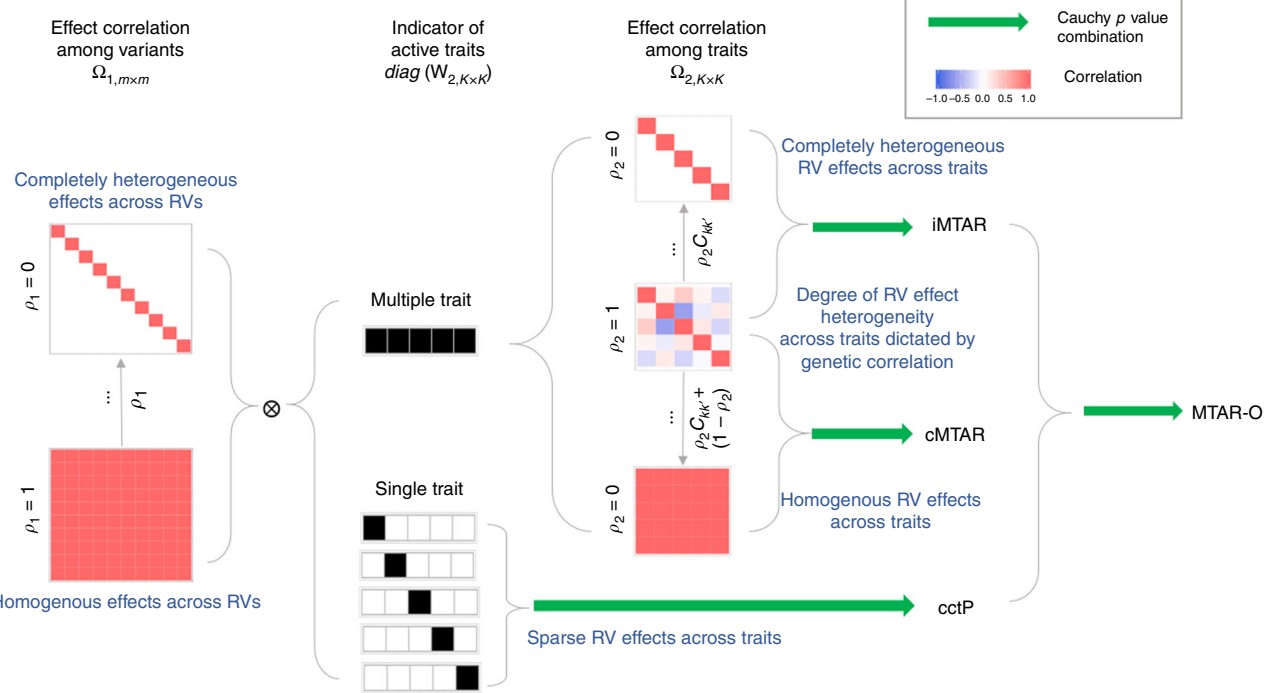

**Fig. 1 Summary of methods under MTAR framework.** In this illustration, the number of variants is $m = 10$ and the number of traits is $K = 5$. The degree of heterogeneity of among-variant effects is controlled by $\rho_1$. MTAR methods are robust to various patterns of genetic effects across variants by combining variance-component test $P$-values from different specifications of $\rho_1$. The degree of heterogeneity of among-trait effects is controlled by $\rho_2$. By changing the value of $\rho_2$, the degree of heterogeneity of among-trait effects can be weakly, moderately, or strongly dictated by genetic correlation $C_{kk'}$. By setting $\rho_2 = 0$, iMTAR and cMTAR structures assume genetic effects become completely heterogeneous and homogeneous, respectively. MTAR methods are robust to various patterns of genetic effects across traits by combining variance-component test $P$-values from different specifications of $\rho_2$. The cctP that combines the single-trait burden and SKAT tests $P$-values is particularly powerful when only a small number of traits are associated with the set of rare variants. The omnibus test MTAR-O that combines iMTAR, cMTAR, and cctP is robust to all the aforementioned patterns of genetic effects across traits and variants.

Although both $\Sigma$ and **B** are covariance matrices among traits and variants, it is important to note the difference. Matrix $\Sigma$ reflects the correlation due to the residual relatedness among traits in the presence of sample overlap and linkage disequilibrium (LD) among variants. An inaccurate estimate of $\Sigma$ yields inflated type I error in the association testing. On the other hand, the matrix $\mathbf{B} = \mathbf{B}_2 \otimes \mathbf{B}_1$ reflects the similarity of the true gene-level rare-variant effects among traits and variants. This information is unknown a priori; hence, **B** needs to be pre-specified. The power of the tests can be greatly improved if the specification reflects the truth. MTAR utilizes the genetic correlation, a global measure of cross-trait genetic similarity, to guide the specification of $\mathbf{B}_2$. The effectiveness of this strategy in gaining power has been demonstrated in the following sections.

**Application of MTAR to GLGC**. We performed multi-trait RVAS for three plasma lipid traits: LDL, HDL, and TG. The GLGC data set includes ~300,000 individuals of primarily European ancestry genotyped with the HumanExome BeadChip (exome array)[33]. The participants were from 73 different studies and single-variant association summary statistics were combined across studies via fixed-effects meta-analysis[32]. The acquisition of the GLGC summary statistics is described in Methods.

Following Liu et al.[33], we considered 179,884 rare variants with minor allele frequency (MAF) < 5% and the highest priority according to their functionality and deleteriousness. We focused on 15,378 genes that contain at least two rare variants. In our analysis, we used the previously reported genetic correlation estimates among the three lipid traits[27] in MTAR. Specifically, the genetic correlation is −0.61 for the pair (HDL, TG), 0.35 for (LDL, TG), and 0.09 for (LDL, HDL). For comparison, we performed the single-trait-based analysis by combining SKAT and burden test $P$-values across traits using either the cctP or the Bonferroni-corrected minimal $P$-value (minP, take the minimal $P$-values and then multiply it by the number of tests combined).

Similar to the previous gene-based RVAS of GLGC data[33], the slightly elevated genomic control lambdas in the quantile–quantile plots suggest the polygenic inheritance of the lipid traits (Supplementary Fig. 1). At a significance threshold of $P < 3.3 \times 10^{-6}$ (corresponding to 0.05/15,378), a total of 140 genes were identified by at least one test (Supplementary Table 2). MTAR tests (MTAR-O, cMTAR, iMTAR) identified 139 genes and the single-trait-based tests (cctP and minP) identified 99 genes (Fig. 2). There are 41 genes exclusively identified by MTAR tests and the MTAR $P$-values for many of these genes are 100-fold smaller than the single-trait-based $P$-values (Table 1, Manhattan plots in Fig. 3 and Supplementary Fig. 2). There is only one gene (*HFE*, Supplementary Table 2) missed by MTAR but its MTAR-O $P$-value ($4.8 \times 10^{-6}$) is close to the single-trait-based $P$-values ($1.8 \times 10^{-6}$).

Most discovered genes (>60%) have the smallest $P$-value when $\rho_2$ is large ($\rho_2 \geq 0.5$), highlighting the informativeness of using genetic correlations to guide the among-trait effect correlation (Supplementary Fig. 3). For those genes, the association patterns among traits are generally consistent with their genetic correlations: genetic effects on HDL and TG are negatively correlated and effects on LDL and TG are positively correlated (Fig. 4a). The cMTAR and iMTAR tests produce similar $P$-values in this case. About 18% of the discovered genes become insignificant if we do not use genetic correlations and simply assume the exchangeable correlation structure in $\mathbf{B}_2$.

When the effects between-trait are strongly heterogeneous and vary randomly among traits (Fig. 4b), iMTAR produces much smaller $P$-values than other tests. When the effects between-trait effects are largely homogeneous (Fig. 4c), cMTAR provides the

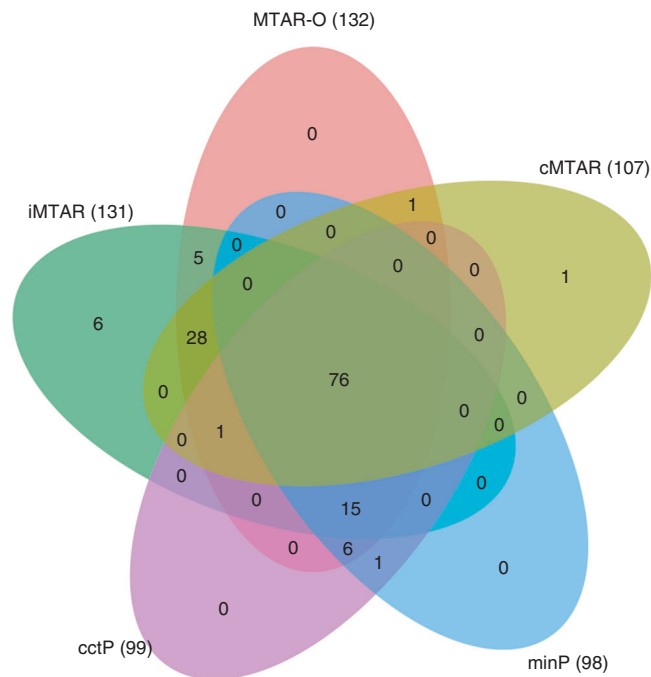

**Fig. 2 Venn diagram of significant genes in the GLGC data analysis.** MTAR-O, cMTAR, iMTAR, cctP, and minP test are performed and the number of significant genes identified by each method is shown in the parentheses.

strongest evidence of association. When the gene is associated with one trait, the single-trait-based analysis (cctP and minP) is desirable (Fig. 4d). MTAR-O has the $P$-value close to the smallest $P$-values among all tests in all the identified genes (Supplementary Table 2).

Many of the 139 MTAR identified genes have an established role in the three lipid traits, including targets for LDL lowering drugs (e.g., *PCSK9*, *NPC1L1*, and *PPARA*) and genes with known association with lipid-related Mendelian disorders (e.g., *LDLR*, *ABCG5*, *APOB*, *ABCA1*, *LCAT*, *APOA1*, and *CETP*). Gene set enrichment analysis of the 139 genes highlighted the gene sets related to lipid metabolism and transport (Supplementary Fig. 4 and Supplementary Data 1), similar to the reported findings from gene set enrichment analysis of GWAS loci for LDL, HDL, and TG[34]. Tissue enrichment analysis of all 139 significant genes using either Human Protein Atlas (HPA) or Genotype-Tissue Expression (GTEx) as reference sets demonstrated enrichment of liver-specific genes (Supplementary Fig. 5), in accordance with a published tissue eQTL enrichment analysis across GWAS loci associated with LDL, HDL, TG, or total cholesterol[35].

Among the 41 genes exclusively identified by MTAR tests, 27 (66%) genes have previously reported association evidence with at least one of the three lipid traits and 20 (74%) of them are associated with at least two lipid traits (Table 1). To replicate the associations of the genes without any existing annotation evidence, we applied the MTAR-O test to an independent UK Biobank GWAS data (Methods). Despite the fact that UK Biobank GWAS data usually harbor a smaller number of rare variants in a gene than GLGC exome chip data, 7 out of 11 (64%) genes were found significant in the UK Biobank at $\alpha = 0.05/11 = 4.5 \times 10^{-3}$ (Table 1 and Supplementary Table 3). These seven validated MTAR discovered genes may have causal impact on the lipid traits. One example is *PNPLA2*, which encodes the enzyme adipose TG lipase (ATGL); ATGL is involved in the breakdown of TG. Although variants associated with *PNPLA2* have not previously been directly linked with any of the three lipid traits in

**Table 1 Results for the 41 genes exclusively identified by MTAR tests in the GLGC analysis.**

| Chr. | Gene | GLGC | | | | | | UKB Neale v2 | | Annotation |
|---|---|---|---|---|---|---|---|---|---|---|
| | | Size | MTAR-O | cMTAR | iMTAR | cctP | minP | Size | MTAR-O | |
| 1 | *COL24A1* | 43 | $2\times10^{-7}$ | $1\times10^{-7}$ | $8\times10^{-8}$ | $7\times10^{-5}$ | $1\times10^{-4}$ | 23 | $5\times10^{-1}$ | |
| 1 | *MCL1* | 6 | $2\times10^{-8}$ | $3\times10^{-7}$ | $8\times10^{-9}$ | $1\times10^{-5}$ | $1\times10^{-5}$ | | | LDL, HDL |
| 2 | ***ASB3\|GPR75-ASB3*** | 14 | $4\times10^{-9}$ | $1\times10^{-4}$ | $1\times10^{-9}$ | $4\times10^{-6}$ | $4\times10^{-6}$ | 5 | $2\times10^{-3}$ | |
| 3 | *ITIH3* | 24 | $3\times10^{-6}$ | $8\times10^{-6}$ | $9\times10^{-7}$ | $1\times10^{-3}$ | $2\times10^{-3}$ | | | HDL |
| 3 | *STAB1* | 62 | $1\times10^{-11}$ | $1\times10^{-11}$ | $5\times10^{-12}$ | $6\times10^{-6}$ | $7\times10^{-6}$ | | | LDL, HDL, TG |
| 4 | *MTTP* | 16 | $3\times10^{-6}$ | $1\times10^{-6}$ | $4\times10^{-5}$ | $7\times10^{-4}$ | $7\times10^{-4}$ | | | LDL HDL, TG |
| 4 | *PLA2G12A* | 2 | $9\times10^{-9}$ | $8\times10^{-9}$ | $5\times10^{-9}$ | $4\times10^{-6}$ | $6\times10^{-6}$ | | | HDL, TG |
| 5 | ***SPARC*** | 6 | $3\times10^{-6}$ | $3\times10^{-6}$ | $2\times10^{-6}$ | $3\times10^{-4}$ | $3\times10^{-4}$ | 5 | $2\times10^{-3}$ | |
| 6 | ***C2*** | 17 | $6\times10^{-10}$ | $9\times10^{-10}$ | $3\times10^{-10}$ | $2\times10^{-5}$ | $3\times10^{-5}$ | 7 | $1\times10^{-3}$ | |
| 6 | *C6orf10* | 9 | $1\times10^{-6}$ | $1\times10^{-6}$ | $7\times10^{-7}$ | $9\times10^{-5}$ | $9\times10^{-5}$ | | | TG |
| 6 | *HLA-DQB1* | 5 | $5\times10^{-8}$ | $5\times10^{-8}$ | $3\times10^{-8}$ | $4\times10^{-4}$ | $5\times10^{-4}$ | | | LDL, TG |
| 6 | *NOTCH4* | 37 | $2\times10^{-8}$ | $5\times10^{-4}$ | $5\times10^{-9}$ | $6\times10^{-5}$ | $8\times10^{-5}$ | | | LDL, TG |
| 6 | *ZNF76* | 22 | $8\times10^{-7}$ | $3\times10^{-7}$ | $2\times10^{-5}$ | $2\times10^{-3}$ | $3\times10^{-3}$ | | | LDL, HDL, TG |
| 7 | *KIAA1324L* | 12 | $2\times10^{-8}$ | $2\times10^{-8}$ | $1\times10^{-8}$ | $3\times10^{-5}$ | $3\times10^{-5}$ | | | LDL |
| 8 | *ZNF572* | 15 | $1\times10^{-6}$ | $2\times10^{-6}$ | $5\times10^{-7}$ | $1\times10^{-4}$ | $2\times10^{-4}$ | | | LDL, HDL, TG |
| 11 | *CKAP5* | 16 | $5\times10^{-6}$ | $4\times10^{-6}$ | $3\times10^{-6}$ | $7\times10^{-4}$ | $7\times10^{-4}$ | | | HDL, TG |
| 11 | *CREB3L1* | 9 | $2\times10^{-7}$ | $2\times10^{-7}$ | $1\times10^{-7}$ | $6\times10^{-6}$ | $6\times10^{-6}$ | | | LDL, HDL, TG |
| 11 | *DSCAML1* | 25 | $1\times10^{-6}$ | $2\times10^{-6}$ | $4\times10^{-7}$ | $3\times10^{-5}$ | $3\times10^{-5}$ | | | LDL, HDL, TG |
| 11 | *MEN1* | 4 | $4\times10^{-6}$ | $4\times10^{-6}$ | $2\times10^{-6}$ | $5\times10^{-5}$ | $7\times10^{-5}$ | | | TG |
| 11 | *NR1H3* | 7 | $4\times10^{-7}$ | $4\times10^{-7}$ | $2\times10^{-7}$ | $3\times10^{-5}$ | $3\times10^{-5}$ | | | LDL, HDL, TG |
| 11 | ***OR8U1\|OR8U8*** | 7 | $3\times10^{-6}$ | $3\times10^{-6}$ | $2\times10^{-6}$ | $6\times10^{-4}$ | $1\times10^{-3}$ | 4 | $2\times10^{-11}$ | |
| 11 | *PLCB3* | 14 | $3\times10^{-9}$ | $3\times10^{-9}$ | $1\times10^{-9}$ | $5\times10^{-5}$ | $8\times10^{-5}$ | | | HDL, TG |
| 11 | ***PNPLA2*** | 14 | $5\times10^{-8}$ | $5\times10^{-8}$ | $3\times10^{-8}$ | $3\times10^{-6}$ | $3\times10^{-6}$ | 5 | $6\times10^{-8}$ | |
| 11 | *SIDT2* | 21 | $5\times10^{-6}$ | $7\times10^{-6}$ | $3\times10^{-6}$ | $8\times10^{-6}$ | $8\times10^{-6}$ | | | LDL, HDL, TG |
| 11 | *TSGA10IP* | 15 | $7\times10^{-9}$ | $8\times10^{-9}$ | $3\times10^{-9}$ | $2\times10^{-4}$ | $3\times10^{-4}$ | | | HDL, TG |
| 12 | *ACADS* | 8 | $4\times10^{-6}$ | $2\times10^{-5}$ | $2\times10^{-6}$ | $4\times10^{-6}$ | $6\times10^{-6}$ | | | LDL, HDL |
| 12 | *ACVRL1* | 8 | $5\times10^{-6}$ | $5\times10^{-6}$ | $3\times10^{-6}$ | $6\times10^{-4}$ | $1\times10^{-3}$ | 5 | $2\times10^{-2}$ | |
| 12 | *C12orf41* | 4 | $1\times10^{-6}$ | $1\times10^{-6}$ | $6\times10^{-7}$ | $8\times10^{-5}$ | $1\times10^{-4}$ | 2 | $3\times10^{-2}$ | |
| 12 | *CMAS* | 3 | $8\times10^{-7}$ | $7\times10^{-7}$ | $4\times10^{-7}$ | $4\times10^{-5}$ | $4\times10^{-5}$ | | | |
| 12 | *SH2B3* | 15 | $2\times10^{-6}$ | $2\times10^{-6}$ | $1\times10^{-6}$ | $5\times10^{-5}$ | $5\times10^{-5}$ | | | LDL, HDL, TG |
| 14 | *DDHD1* | 9 | $7\times10^{-7}$ | $6\times10^{-7}$ | $3\times10^{-7}$ | $1\times10^{-4}$ | $2\times10^{-4}$ | | | |
| 14 | *PCK2* | 34 | $2\times10^{-6}$ | $3\times10^{-6}$ | $1\times10^{-6}$ | $4\times10^{-4}$ | $6\times10^{-4}$ | | | LDL |
| 15 | ***ARRDC4*** | 8 | $5\times10^{-6}$ | $5\times10^{-6}$ | $3\times10^{-6}$ | $7\times10^{-4}$ | $2\times10^{-3}$ | 4 | $1\times10^{-4}$ | |
| 16 | *CFDP1* | 7 | $2\times10^{-6}$ | $1\times10^{-6}$ | $8\times10^{-7}$ | $9\times10^{-4}$ | $1\times10^{-3}$ | 4 | $2\times10^{-2}$ | |
| 17 | *BECN1* | 4 | $2\times10^{-6}$ | $5\times10^{-6}$ | $1\times10^{-6}$ | $1\times10^{-3}$ | $2\times10^{-3}$ | | | |
| 17 | *GEMIN4* | 35 | $2\times10^{-6}$ | $7\times10^{-4}$ | $5\times10^{-7}$ | $4\times10^{-4}$ | $6\times10^{-4}$ | | | HDL |
| 17 | *SHBG* | 7 | $3\times10^{-6}$ | $3\times10^{-6}$ | $1\times10^{-6}$ | $5\times10^{-6}$ | $6\times10^{-6}$ | | | LDL, TG |
| 19 | ***AXL*** | 11 | $1\times10^{-6}$ | $1\times10^{-6}$ | $7\times10^{-7}$ | $2\times10^{-4}$ | $2\times10^{-4}$ | 3 | $3\times10^{-5}$ | |
| 19 | *LAIR1* | 13 | $3\times10^{-7}$ | $2\times10^{-7}$ | $1\times10^{-7}$ | $8\times10^{-5}$ | $1\times10^{-4}$ | | | LDL, HDL |
| 19 | *LOC55908* | 7 | $8\times10^{-10}$ | $9\times10^{-10}$ | $4\times10^{-10}$ | $9\times10^{-6}$ | $1\times10^{-5}$ | | | HDL, TG |
| 21 | *COL18A1* | 44 | $6\times10^{-8}$ | $4\times10^{-8}$ | $4\times10^{-8}$ | $7\times10^{-5}$ | $7\times10^{-5}$ | | | TG |

Seven novel genes replicated in the UK Biobank analysis are shown in bold.
The annotation is the summary of association evidence from the Open Targets[52,53] and the STOPGAP[54] databases, and the previous analysis of the GLGC data[33] for the three traits.
The genes (*BECN1*, *CAMS*, and *DDHD1*) with cumulative minor allele counts <10 in the UK Biobank are not analyzed in the replication stage.

humans, ATGL-knockout mice display altered very-low-density lipoprotein, HDL, and TG levels[36].

**Simulation studies**. We used simulation studies to further investigate the type I error control and power of MTAR. We considered three continuous traits that have similar residual covariances as the three lipid traits in the real data[37]. We simulated data in three cohorts ($N_1 = 3000$, $N_2 = 3500$, $N_3 = 2000$) that have different patterns of sample overlap for the three traits (Supplementary Fig. 6). The details of genotype and phenotype simulations are provided in Methods.

As in the GLGC data analysis, we utilized the combined summary statistics across three cohorts for each trait (Methods). We first evaluated the empirical type I error rates based on $10^8$ replicates of simulation. Prior research has shown that the

accuracy of the Cauchy combined P-value is generally satisfactory for practical use in rare-variant association tests, but a slight inflation is possible[30]. Reassured that type I error was well controlled (Supplementary Table 4), we then proceeded to simulate traits under the alternative model to evaluate power. The percentage of causal variants was set to be 50% or 20% for scenarios of dense and sparse signals. For the causal variant $j$ in trait $k$, the genetic effect was set to $\beta_{kj} = s_j^{\mathrm{snp}} \times s_k^{\mathrm{trait}} \times d|\log_{10}\mathrm{MAF}_j|$, where $s_j^{\mathrm{snp}}$ and $s_k^{\mathrm{trait}}$ determined the heterogeneity of the effect directions among variants and traits, respectively, and $d|\log_{10}\mathrm{MAF}_j|$ stated that the effect size was larger for the variant with smaller MAF. We set different values of $d$ for different percentage of causal and $s_j^{\mathrm{snp}}$ settings such that the power of the tests in each setting is reasonably high. The effects among causal variants are either in the same direction

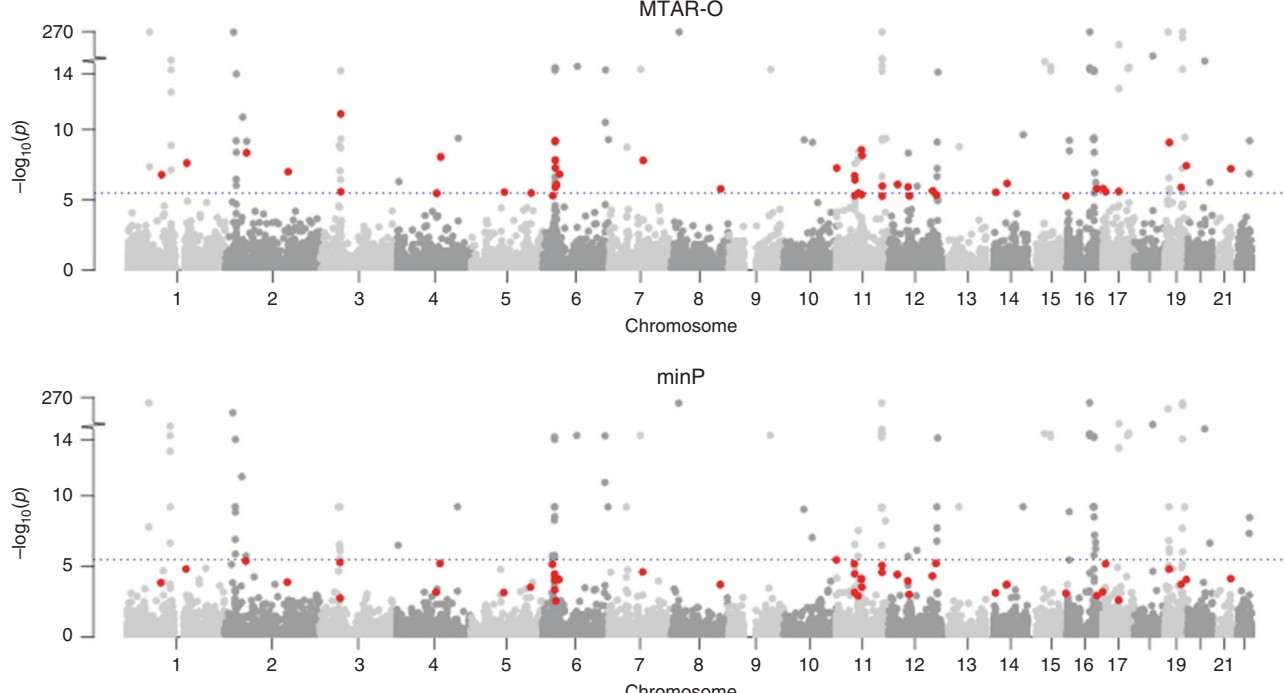

**Fig. 3 Manhattan plots of MTAR-O and minP results in the GLGC data analysis.** The horizontal line marks the genome-wide significance threshold ($3.3 \times 10^{-6}$). The 41 genes highlighted in red are those exclusively discovered by MTAR tests (MTAR-O, cMTAR, and iMTAR). The Manhattan plots for the other methods (cMTAR, iMTAR, and cctP) are shown in Supplementary Fig. 2.

($s_j^{\text{snp}} = 1$ for all $j$) or bidirectional (randomly assign 1 or −1 with equal probability to $s_j^{\text{snp}}$). To run MTAR, we utilized the genetic correlations in GLGC data analysis for LDL, HDL, and TG, but we did not specify $s_k^{\text{trait}}$ according to their genetic correlations. In particular, we considered five patterns of $s_k^{\text{trait}}$ across traits: $(s_1^{\text{trait}}, s_2^{\text{trait}}, s_3^{\text{trait}}) = (0, 0, 1); (0, 1, 1); (0, -1, 1); (1, -1, 1)$ and $(1, 1, 1)$. All the association patterns across traits and variants considered in our power simulation are visualized in Supplementary Fig. 7.

The empirical power is estimated at the significance level of $\alpha = 2.5 \times 10^{-6}$ based on $10^4$ replicates (Fig. 5). When the gene is associated with one trait (pattern 1 of $s_k^{\text{trait}}$), the single-trait-based tests (cctP and minP) are more powerful than iMTAR and cMTAR but the trend is reversed in other patterns. cMTAR is more powerful than iMTAR when the effects are homogeneous (pattern 5). iMTAR is much more powerful than cMTAR when the effects are heterogeneous and the specified genetic correlations are not informative to the true relationship of the effects among traits (pattern 2). The power of MTAR-O is close to the most powerful test in all scenarios. These observations are consistent with results from the GLGC data analysis.

**Comparison with other multi-trait multi-variant methods.** In comparison with existing multi-trait multi-variant methods (Supplementary Table 1), MTAR has a unique combination of features that make it desirable for practical use. First, MTAR uses summary statistics rather than individual-level data. MTAR starts with simple summary statistics calculated in a study for each trait: variant-level score statistics and their covariance estimates[24]. These statistics can be easily constructed using the information routinely shared in public domains[38]. Compared with methods that require pooling individual-level data, using summary statistics can better protect study participant privacy and reduce logistical difficulties and computational burden. Second, MTAR

allows the summary statistics for different traits to come from (possibly unknown) overlapping samples. Failure to account for the correlation between summary statistics induced by the over-lapping samples can greatly inflate type I error[39]. Sample overlap is prevalent in the multi-trait analysis. Sometimes the overlap pattern is clear (e.g., all traits are measured in the same study or in different studies that share controls[40,41]), but other times is often elusive—public domains only have combined summary statistics across many studies for each trait and study-specific summary statistics are not available[7]. MTAR can handle these scenarios and use a simple approach to accurately estimate the correlation between summary statistics for the traits with sample overlap. Third, MTAR is computationally fast. The MTAR $P$-value calculation is analytical and does not require time-consuming procedures such as permutation and Monte Carlo simulation.

We compared the power of MTAR with Multi-SKAT[10] (MultiSKAT R package) and MTaSPUsSet[17] (aSPU R package) in numerical studies. These two existing methods have demonstrated superior performance to other competing multi-trait multi-variant methods such as metaCCA[18], MGAS[19], DKAT[11], MAAUSS[13], MSKAT[15], and GAMuT[14]. Similar to MTAR, Multi-SKAT and MTaSPUsSet proposed several tests to accommodate different patterns of associations across traits and variants, and omnibus tests to gain robustness. We compared their omnibus tests with MTAR-O. In the simulation study, we let all cohorts have complete trait values as it is required by Multi-SKAT. Empirical power was estimated at the $\alpha = 10^{-4}$ level due to the speed of MTaSPUsSet. MTAR-O has greater power than Multi-SKAT and MTaSPUsSet in almost all scenarios, especially when the genetic correlation reflects the heterogeneity of effects among traits (patterns 3–4 of $s_k^{\text{trait}}$) (Supplementary Fig. 8). Furthermore, MTAR-O is computationally more efficient. Multi-SKAT and MTaSPUsSet, respectively, take 29 and 184 s on average to complete one replicate of simulation, whereas MTAR-O only

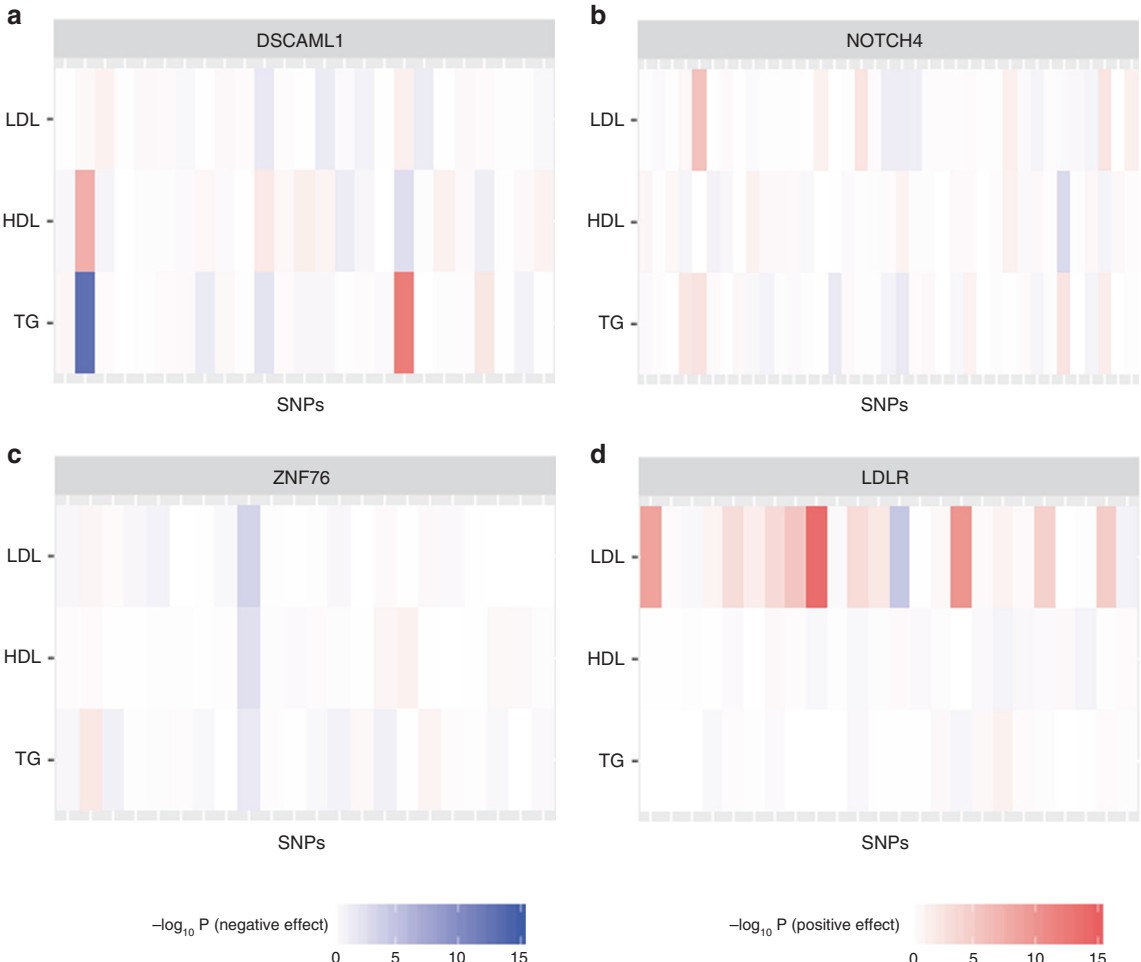

**Fig. 4 Heat maps of association signals in the GLGC data for four example genes.** The darkness of the color indicates the variant-level *Z*-test *P*-values (in $-\log_{10}$ scale) for individual traits LDL, HDL, and TG. The positive and negative *Z*-scores are indicated by red and blue colors, respectively. **a** The effect correlations among traits resemble their genetic correlations. **b** The effects are independent among traits. **c** The effects are similar among traits. **d** Association signal resides in a single trait.

takes 10 s. In addition, we applied MTaSPUsSet that does not require individual-level data to the GLGC summary statistics. MTaSPUsSet missed 52 MTAR identified genes, whereas MTAR only missed 9 MTaSPUsSet identified genes.

## Discussion

We have introduced MTAR, a framework for conducting the meta-analysis of RVAS summary statistics across multiple traits. The cMTAR, iMTAR, and cctP tests cover a wide variety of association patterns among traits and variants. The omnibus test MTAR-O achieves robust and high power by combining the *P*-values of the three complementary tests. The use of summary statistics and Cauchy *P*-value combination method empowers MTAR to conduct whole-genome multi-trait RVAS in a computationally efficient manner. The computation time of running MTAR methods on the simulated and GLGC datasets are summarized in Methods. Our numerical results have confirmed that MTAR tests properly control the type I error in the present of complex patterns of sample overlap among traits and have substantial power gain relative to the separate analysis of RVAS for each trait. In the analysis of lipid traits in GLGC, MTAR identified many more genes than single-trait-based tests, including genes that have not been previously linked to lipid traits and

represent novel findings. Many of these genes have been successfully replicated in an independent UK Biobank data.

Utilizing genetic correlations to guide the specification of gene-level effects heterogeneity across traits is one main innovation of MTAR. The genetic correlation is a genome-wide measure of the shared genetic architecture between a pair of traits and it is calculated using common variants across the genome. The GLGC data analysis results suggest that the rare-variant effect correlation among traits is generally in accordance with the genetic correlation for most genes and the use of genetic correlation in MTAR helps to substantially improve the power of the multi-trait analysis.

Although we mainly demonstrate MTAR in the analysis of continuous traits, the method can be applied to binary traits (Methods) as long as the score statistics from the models are unbiased and their covariance estimates are accurate. For binary traits, the normal approximation to the score statistics could be inaccurate in the unbalanced case-control setting[42], which could affect the performance of the multi-trait analysis. For studies with related subjects, one may use methods based on mixed models to generate appropriate score statistics[43,44]. Future research is required on how to properly handle various patterns of sample overlap across traits in the presence of familial and cryptic relatedness.

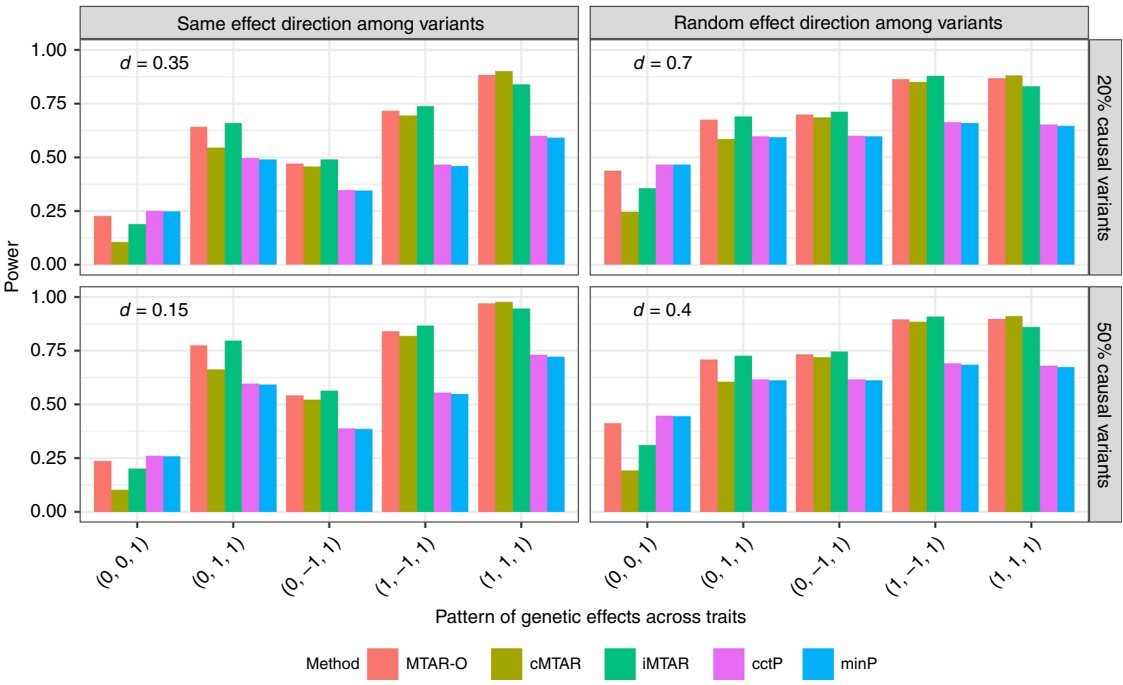

**Fig. 5 Power comparisons of MTAR-O, cMTAR, iMTAR, cctP, and minP.** Each bar represents the empirical power for a method estimated as the proportion of $P$-values $< 2.5 \times 10^{-6}$ based on $10^4$ replicates. The percentage of causal variants is set to be 20% or 50%, which corresponds to the two rows. The left column assumes the effects of the causal variants have the same direction, whereas the right column assumes the effect directions are randomly determined with an equal probability. The effect sizes ($|\beta_{kj}|$'s) of the causal variants have a decreasing relationship with MAF as $|\beta_{kj}| = d|\log_{10} MAF_j|$, where the constant $d$ depends on the percentage of causal variants and the direction of their effects (the value of $d$ is presented in each subfigure). For each configuration in a subfigure, five patterns of among-trait effects are considered.

With the increasing number of complex traits available in large-scale whole exome/genome sequencing studies and electronic health record linked biobank data, multi-trait analysis based on summary statistics of multiple rare variants will become an important tool to boost the power of discovering genetic components of complex traits and unravel their shared genetic architectures. We envision that MTAR will facilitate the accumulation of adequately large sample sizes to accelerate discoveries in complex trait genetics and provide new biological insights by revealing pleiotropic genes.

## Methods

**Covariance of genetic effects among variants.** $\mathbf{B}_1$ is a $m \times m$ covariance matrix for the effects among variants. We set $\mathbf{B}_1 = \mathbf{W}_1 \boldsymbol{\Omega}_1 \mathbf{W}_1$, where $\mathbf{W}_1$ is a diagonal matrix with each element being a variant-specific weight and $\boldsymbol{\Omega}_1$ is a between-variant effect correlation matrix of exchangeable structure with correlation coefficient $\rho_1$ ($0 \le \rho_1 \le 1$). Specifically, the single-trait analysis becomes SKAT[20] if $\rho_1 = 0$ and burden tests[22,24,45,46] if $\rho_1 = 1$. Burden tests are more powerful when the association effects are similar across the aggregated variants, whereas SKAT is more powerful when the effects are in opposite directions or the number of causal variants is small relative to neutral variants. As for the variant-specific weights (in $\mathbf{W}_1$), by default, we set them based on the MAF through a beta distribution density function $Beta(\text{MAF}; 1, 25)$ as in SKAT. Other weighting schemes can be employed as well.

**Summary statistics.** For each trait $k$ ($k = 1, \ldots, K$) and subject $i$ ($i = 1, \ldots, n$), when the individual-level phenotype ($Y_{ik}$), genotypes ($\mathbf{G}_{ik}$), and covariates ($\mathbf{X}_{ik}$) are available, the score statistics $\mathbf{U}_k$ and their covariance $\mathbf{V}_k$ can be obtained from the generalized linear model with the likelihood function $\exp\left\{ \frac{Y_{ik}(\boldsymbol{\beta}_k^T \mathbf{G}_{ik} + \boldsymbol{\gamma}_k^T \mathbf{X}_{ik}) - b(\boldsymbol{\beta}_k^T \mathbf{G}_{ik} + \boldsymbol{\gamma}_k^T \mathbf{X}_{ik})}{a(\phi_k)} + c(Y_{ik}, \phi_k) \right\}$, where $\boldsymbol{\beta}_k$ and $\boldsymbol{\gamma}_k$ are regression parameters, $\phi_k$ is a dispersion parameter, and $a$, $b$, and $c$ are specific functions. Specifically, we have $\mathbf{U}_k = a(\hat{\phi}_k)^{-1} \sum_{i=1}^n \{ Y_{ik} - b'(\hat{\boldsymbol{\gamma}}_k^T \mathbf{X}_{ik}) \} \mathbf{G}_{ik}$ and $\mathbf{V}_k = a(\hat{\phi}_k)^{-1} [\sum_{i=1}^n b''(\hat{\boldsymbol{\gamma}}_k^T \mathbf{X}_{ik}) \mathbf{G}_{ik} \mathbf{G}_{ik}^T - \{\sum_{i=1}^n b''(\hat{\boldsymbol{\gamma}}_k^T \mathbf{X}_{ik}) \mathbf{G}_{ik} \mathbf{X}_{ik}^T\} \{\sum_{i=1}^n b''(\hat{\boldsymbol{\gamma}}_k^T \mathbf{X}_{ik}) \mathbf{X}_{ik} \mathbf{X}_{ik}^T\}^{-1} \{\sum_{i=1}^n b''(\hat{\boldsymbol{\gamma}}_k^T \mathbf{X}_{ik}) \mathbf{X}_{ik} \mathbf{G}_{ik}^T\}]$, where $\hat{\boldsymbol{\gamma}}_k$ and $\hat{\phi}_k$ are the restricted maximum likelihood estimators of $\boldsymbol{\gamma}_k$ and $\phi_k$ under $H_0: \boldsymbol{\beta}_k = 0$, and $b'$ and $b''$ are the first and second derivatives of function $b$. For the linear regression model, we have

$a(\hat{\phi}_k) = n^{-1} \sum_{i=1}^n (Y_{ik} - \hat{\boldsymbol{\gamma}}_k^T \mathbf{X}_{ik})^2$, $b'(z) = z$, and $b''(z) = 1$. For the logistic regression model, we have $a(\hat{\phi}_k) = 1$, $b'(z) = e^z/(1 + e^z)$, and $b''(z) = e^z/(1 + e^z)^2$.

The $\mathbf{U}_k$ and $\mathbf{V}_k$ can also be derived from different forms of summary statistics shared in public domains[38]. When the score statistics $\mathbf{U}_k$ and their variances (i.e., $\text{diag}(\mathbf{V}_k)$) are available, the covariance matrix of $\mathbf{U}_k$ can be approximated as $\mathbf{V}_k \approx \{\text{diag}(\mathbf{V}_k)\}^{1/2} \mathbf{R} \{\text{diag}(\mathbf{V}_k)\}^{1/2}$, where $\mathbf{R} = \{R_{j\ell}\}_{j,\ell=1}^m$ is the SNP LD matrix calculated from the Pearson correlation coefficient among the genotypes of the $m$ variants based on the working genotypes or external reference. In another case, when the effect estimates $\hat{\boldsymbol{\beta}}_k = \{\hat{\beta}_{kj}\}_{j=1}^m$ and their standard errors $\mathbf{se}_k = \{se_{kj}\}_{j=1}^m$ are available, we can approximate $\mathbf{U}_k = \{U_{kj}\}_{j=1}^m$ and $\mathbf{V}_k = \{V_{kj\ell}\}_{j,\ell=1}^m$ as $U_{kj} \approx \hat{\beta}_{kj}/se_{kj}^2$ and $V_{kj\ell} \approx R_{j\ell}/(se_{kj} se_{k\ell})$.

**Covariance of summary statistics between traits.** If all the traits are from the same study or multiple studies with overlapping samples, the summary statistics $\mathbf{U}_k$ among traits $k = 1, \ldots, K$ are correlated. Assume trait $k$ is from cohort $A$ with sample size $n_A$ and trait $k'$ is from cohort $B$ with sample size $n_B$, and there are $n_C$ overlapping subjects in these two cohorts. For any SNP $j$ not associated with the traits, the correlation matrix of $Z$-score $U_{kj}/\sqrt{V_{kj}}$ among traits is invariant to SNP $j$[39,47]. In particular, if both traits $k$ and $k'$ are quantitative, we have

$$\zeta_{kk'} \equiv \text{cov}\left( \frac{U_{kj}}{\sqrt{V_{kj}}}, \frac{U_{k'j}}{\sqrt{V_{k'j}}} \right) \approx \frac{n_C}{\sqrt{n_A}\sqrt{n_B}} \text{cor}(Y_k, Y_{k'}). \quad (1)$$

If both traits $k$ and $k'$ are binary, let $n_{C0}$ ($n_{C1}$) represent the number of overlapping samples with trait value of 0 (or 1), $n_{A0}$ ($n_{A1}$) denotes the number of subjects with trait $k$ and takes the value of 0 (or 1) and $n_{B0}$ ($n_{B1}$) denotes the number of subjects with trait $k'$ and takes the value of 0 (or 1), then we have[39]

$$\zeta_{kk'} \equiv \text{cov}\left( \frac{U_{kj}}{\sqrt{V_{kj}}}, \frac{U_{k'j}}{\sqrt{V_{k'j}}} \right) \approx \left( n_{C0}\sqrt{\frac{n_{A1}n_{B1}}{n_{A0}n_{B0}}} + n_{C1}\sqrt{\frac{n_{A0}n_{B0}}{n_{A1}n_{B1}}} \right) / \sqrt{n_A n_B}. \quad (2)$$

Hence, we can accurately estimate $\zeta_{kk'}$ using the independent null variants across the whole genome. Specifically, we first perform LD pruning using LD threshold $r^2 < 0.01$ in 500 kb region to obtain a set of independent common variants. We then remove variants with association test $P$-values $< 0.05$ and only keep variants that are not associated with any traits. For any traits $k$ and $k'$, we calculate the between-trait sample correlation of the $Z$-scores on the remaining variants and denote it as $\tilde{\zeta}_{kk'}$. In our simulation study, we benchmarked $\tilde{\zeta}_{kk'}$ against empirical sample covariance of $Z$-scores and confirmed the accuracy of the estimate $\tilde{\zeta}_{kk'}$ (Supplementary

Fig. 9). Finally, provided the gene is not associated with any trait, the covariance of $\widehat{\boldsymbol{\beta}}_k$ and $\widehat{\boldsymbol{\beta}}_{k'}$ can be estimated using $\widehat{\zeta}_{kk'}$

$$\mathrm{cov}(\widehat{\boldsymbol{\beta}}_k, \widehat{\boldsymbol{\beta}}_{k'}) = \mathrm{cov}(\mathbf{V}_k^{-1}\mathbf{U}_k, \mathbf{V}_{k'}^{-1}\mathbf{U}_{k'}) \approx \widehat{\zeta}_{kk'} \mathbf{V}_k^{-1}\{\mathrm{diag}(\mathbf{V}_k)\}^{1/2}\mathbf{R}\{\mathrm{diag}(\mathbf{V}_{k'})\}^{1/2}\mathbf{V}_{k'}^{-1}, \quad (3)$$

where the matrix $\mathbf{R}$ is the SNP LD matrix defined in the previous subsection.

**MTAR test statistics and P-values.** We let $\boldsymbol{\beta} = (\boldsymbol{\beta}_1^{\mathrm{T}}, \dots, \boldsymbol{\beta}_K^{\mathrm{T}})^{\mathrm{T}}$ denote the $m$ genetic effects across $K$ traits and $\widehat{\boldsymbol{\beta}} = (\mathbf{V}_1^{-1}\mathbf{U}_1, \dots, \mathbf{V}_K^{-1}\mathbf{U}_K)$ denote their effect estimates constructed from $\mathbf{U}_k$ and $\mathbf{V}_k$. The MTAR framework assumes the hierarchical model

$$\widehat{\boldsymbol{\beta}}|\boldsymbol{\beta} \sim \mathcal{N}(\boldsymbol{\beta}, \boldsymbol{\Sigma}), \quad \boldsymbol{\beta} \sim \mathcal{N}(\mathbf{0}, \sigma\mathbf{B}). \quad (4)$$

As described in the main text, $\boldsymbol{\Sigma}$ reflects the correlation due to the residual relatedness among traits in the presence of sample overlap and LD among variants, and $\mathbf{B}$ reflects the correlation among the rare-variant effects across traits and variants. The $\mathbf{B}$ matrix contains two coefficients $\rho_1$ and $\rho_2$, where $\rho_1$ controls the effect correlation among variants and $\rho_2$ controls the contribution of the genetic correlation to the among-trait rare-variant effect correlation.

For a fixed set of $\rho = (\rho_1, \rho_2)$, we test $H_0 : \sigma = 0$ against $H_1 : \sigma \neq 0$ by a variance-component score test[48]:

$$Q_\rho = \widehat{\boldsymbol{\beta}}^{\mathrm{T}}\boldsymbol{\Sigma}^{-1}\mathbf{B}\boldsymbol{\Sigma}^{-1}\widehat{\boldsymbol{\beta}}. \quad (5)$$

The test statistic follows a mixture of $\chi^2$ distribution under the null hypothesis. Davies method can be used to accurately estimate the $P$-value[49]. In addition, rare variants often show polymorphisms in some but not all traits, the adjustment of the formula for this case is described in the Supplementary Methods.

In the cMTAR and iMTAR tests (respectively correspond to two specifications of effect correlation among traits in $\mathbf{B}$), the Cauchy $P$-value combination method is utilized to combine results from various $\rho_1$ and $\rho_2$. Similar to the minimum $P$-value method, the Cauchy method mainly focuses on a few smallest $P$-values[30]. The advantage of the Cauchy method over the minimum $P$-value method is that the Cauchy method is computationally fast because it does not rely on the Monte Carlo simulation to account for the correlation of the individual tests[29]. Specifically, the iMTAR or cMTAR test statistic is defined as

$$Q_{\mathrm{iMTAR/cMTAR}} = \frac{\sum_{\rho \in \mathcal{S}} \tan\left[\left\{0.5 - p(Q_\rho)\right\}\pi\right]}{|\mathcal{S}|}, \quad (6)$$

where $p(Q_\rho)$ is the $P$-value of $Q_\rho$, $\mathcal{S}$ is a set that includes a grid of possible values of $\rho = (\rho_1, \rho_2)$, and $|\mathcal{S}|$ is the size of the set. In our implementation, we consider the grid $\{0, 0.5, 1\}$ for both $\rho_1$ and $\rho_2$ such that there are nine combinations. We have shown in the Supplementary Fig. 10 that the GLGC analysis results are not sensitive to the choice of the grid. The $P$-value of $Q_{\mathrm{iMTAR/cMTAR}}$ can be accurately approximated by $0.5 - \arctan(Q_{\mathrm{iMTAR/cMTAR}})/\pi$[29].

In addition, the MTAR framework reduces to single-trait analysis when we set a single diagonal element of matrix $\mathbf{W}_2$ to 1 (Fig. 1). We use the Cauchy method to combine these single-trait $P$-values from SKAT and burden tests and construct the cctP test as

$$Q_{\mathrm{cctP}} = \frac{\sum_{k=1}^{K} \tan\left[\left\{0.5 - p_{\mathrm{skat},k}\right\}\pi\right] + \sum_{k=1}^{K} \tan\left[\left\{0.5 - p_{\mathrm{burden},k}\right\}\pi\right]}{2K}, \quad (7)$$

where $p_{\mathrm{skat},k}$ and $p_{\mathrm{burden},k}$ are the $P$-values from the SKAT and burden tests for trait $k$. The $P$-value of the cctP test can be approximated by $0.5 - \arctan(Q_{\mathrm{cctP}})/\pi$.

Finally, the Cauchy method is used to construct MTAR-O test by combining $P$-values from cMTAR, iMTAR, and cctP as

$$Q_{\mathrm{MTAR\text{-}O}} = \frac{\tan[\{0.5 - p_{\mathrm{cMTAR}}\}\pi] + \tan[\{0.5 - p_{\mathrm{iMTAR}}\}\pi] + \tan[\{0.5 - p_{\mathrm{cctP}}\}\pi]}{3}, \quad (8)$$

where $p_{\mathrm{cMTAR}}$, $p_{\mathrm{iMTAR}}$, and $p_{\mathrm{cctP}}$ are the $P$-values of the cMTAR, iMTAR, and cctP tests. The $P$-value of the MTAR-O test can be approximated by $0.5 - \arctan(Q_{\mathrm{MTAR\text{-}O}})/\pi$.

**Summary statistics from the GLGC.** The summary statistics for the lipid traits were downloaded from http://csg.sph.umich.edu/abecasis/public/lipids2017/. For each trait $k$, the web portal contains variant-level genetic effect estimates $\widehat{\boldsymbol{\beta}}_k$ for a given gene and their standard errors $\mathrm{se}_k$. We obtained $\mathbf{U}_k$ and $\mathbf{V}_k$ by using $\widehat{\boldsymbol{\beta}}_k$ and $\mathrm{se}_k$ as described in Summary statistics subsection of Methods. As the original genotypes from the study are not publicly available, we estimated the LD matrix $\mathbf{R}$ based on the genotypes of the European population from the NHLBI Exome Sequencing Project (ESP)[37]. To account for possible sample overlap among traits, we used Eq. (3) to estimate covariance among summary statistics across traits.

**Gene set and tissue enrichment analysis.** Gene set enrichment analysis was conducted using the one-sided hypergeometric test against Reactome Pathways and Gene Ontology Biological Processes, as implemented in the GENE2FUNC from

FUMA[50], with the genes tested in MTAR used as the background gene set. Enrichment $P$-values are adjusted for multiplicity using the Benjamini–Hochberg procedure within each set type tested; sets with adjusted $P$-value less than 0.05 are reported. Tissue enrichment analysis was conducted using TissueEnrich[51], which implements the one-sided hypergeometric test for enrichment of user-defined genes relative to lists of tissue-specific, tissue-enhanced, and group-enhanced genes. Default settings for the definition of tissue-enriched and enhanced genes from both GTEx and HPA RNA-seq datasets were applied. Enrichment $P$-values are adjusted for multiplicity using the Benjamini–Hochberg procedure within each reference set (GTEx and HPA).

**Gene association annotation.** We annotated the 41 genes exclusively discovered by MTAR in the GLGC data analysis using two recently developed databases: Open Targets[52,53] (Supplementary Data 2) and STOPGAP[54] (Supplementary Data 3). Open Targets and STOPGAP both link genes to a trait or disease via annotation of genomic loci detected in GWAS. For each of the 41 genes, the linked diseases are searched and filtered to the three traits: LDL, HDL, and TG, and the variant-disease association $P$-value $< 5 \times 10^{-8}$ from the two databases. In addition, the lipid association results from the Supplementary Tables 9 and 12 in the paper of previous GLGC data analysis[33] were also used to annotate the 41 genes (Supplementary Table 5).

**Replication of significant genes in the UK Biobank data.** To replicate the associations of 14 genes (11 after removing genes with cumulative minor allele counts less than 10 in the UK Biobank GWAS data) identified exclusively by MTAR tests but without any annotation evidence, we applied the MTAR methods to an independent study with association summary statistics from the UK Biobank GWAS data set. The GWAS summary statistics were released by the Neale Lab with the re-release of UK Biobank genotype imputation (termed imputed-v3). The three related traits LDL direct (mmol/L), HDL direct (mmol/L), and TG (mmol/L) were jointly analyzed in a similar manner as the analysis of GLGC data.

**Data simulation.** For all simulations, we generated 100 haplotypes of length 1 MB under a calibrated coalescent model to mimic the LD structure and local combination rate of the European population[55]. These haplotypes were used to form the genotypes of 8500 subjects across three cohorts. To simulate the genotypes for a data set, we randomly selected one thousand 3 KB regions in each haplotype and focused on rare variants with MAF $< 0.05$.

For each subject $i$, three traits were generated based on a multi-response regression model

$$\begin{bmatrix} Y_{i1} \\ Y_{i2} \\ Y_{i3} \end{bmatrix} = \begin{bmatrix} \beta_{11} & \cdots & \beta_{1m} \\ \beta_{21} & \cdots & \beta_{2m} \\ \beta_{31} & \cdots & \beta_{3m} \end{bmatrix} \begin{bmatrix} G_{i1} \\ \vdots \\ G_{im} \end{bmatrix} + 0.1 X_{i1} + 0.2 X_{i2} + \begin{bmatrix} \epsilon_{i1} \\ \epsilon_{i2} \\ \epsilon_{i3} \end{bmatrix}, \quad \begin{bmatrix} \epsilon_{i1} \\ \epsilon_{i2} \\ \epsilon_{i3} \end{bmatrix}$$

$$\sim \mathcal{N}\left(\mathbf{0}, \begin{bmatrix} 1 & 0.1 & 0 \\ 0.1 & 1 & -0.1 \\ 0 & -0.1 & 1 \end{bmatrix}\right), \quad (9)$$

where $\beta_{kj}$ is the genetic effect for trait $k$ at variant $j$, $G_{ij}$ is the genotype at variant $j$, $X_{i1}$ is a binary covariate simulated from Bernoulli(0.5), $X_{i2}$ is a continuous covariate simulated from a standard normal distribution. The covariance matrix of the error term used here is based on the estimated residual correlations among the lipid traits LDL, HDL, and TG in the ESP data[37]. The reduced model was used when we needed to generate only one or two traits for subject $i$.

**Computation time.** We estimated the computation time of MTAR tests by considering different numbers of variants $m = 5, 10, 20, 50, 100$ and traits $K = 3, 6,$ or 9 (Supplementary Fig. 11). For each scenario, we generated 50 datasets and reported the average computation time. On average, MTAR-O, cMTAR, and iMTAR took less than 0.11, 0.06, and 0.05 s (2.4 GHz Intel Core i5, Produced by Intel Co., Santa Clara, CA) when applied to a data set with 20 variants and 3 traits. The computation time did not change much in the presence of sample overlap; but it increased to 1, 0.51, and 0.49 s when the number of traits was increased to 9. MTAR is scalable for genome-wide analysis. Analyzing the GLGC data (15,378 genes) using MTAR-O, cMTAR, and iMTAR took about 25, 10, and 8 h on a laptop with a single core. After the computation jobs were distributed to multiple cores by chromosome, the analysis was finished within 2 h.

**Web resources.** SKAT R package v1.3.2.1: https://cran.r-project.org/web/packages/SKAT MultiSKAT R package v1.0: https://github.com/diptavo/MultiSKAT aSPU R package v1.48: https://cran.r-project.org/web/packages/aSPU FUMA v1.3.5: http://fuma.ctglab.nl TissueEnrich v1.8.0: https://tissueenrich.gdcb.iastate.edu Open Targets: https://genetics.opentargets.org STOPGAP: https://github.com/StatGenPRD/STOPGAP.

**Reporting summary.** Further information on research design is available in the Nature Research Reporting Summary linked to this article.

## Data availability

No data were generated in the present study. The GLGC summary statistics are publicly available at http://csg.sph.umich.edu/abecasis/public/lipids2017/. The UK Biobank GWAS summary statistics data (Neale v2) are described at http://www.nealelab.is/uk-biobank and are publicly available at https://www.dropbox.com/s/2msvdv4axfz362b/30780_raw.gwas.imputed_v3.both_sexes.tsv.bgz?dl=0 for LDL direct (mmol/L); https://www.dropbox.com/s/sn30890f64p0htu/30760_raw.gwas.imputed_v3.both_sexes.tsv.bgz?dl=0 for HDL cholesterol (mmol/L); https://www.dropbox.com/s/0tdxu9g7itbct6m/30870_raw.gwas.imputed_v3.both_sexes.tsv.bgz?dl=0 for triglycerides (mmol/L).

## Code availability

Our method is implemented in the MTAR R package, freely available at the Comprehensive R Archive Network (CRAN): https://cran.r-project.org/web/packages/MTAR.

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

## Acknowledgements

This work was supported by the Data Science Initiative Award provided by the University of Wisconsin-Madison Office of the Chancellor and the Vice Chancellor for Research and Graduate Education with funding from the Wisconsin Alumni Research Foundation. We thank Dr D.J. Liu for providing information on the GLGC data.

## Author contributions

Z.Z.T. and J.S. oversaw the study. The theory underlying MTAR was conceived of and developed by Z.Z.T., with contributions from L.L., J.S., and H.Z. L.L. developed MTAR software and performed lipid data analyses. J.S. and A.C. conducted gene annotation and result interpretation. L.L., J.S., and H.Z. performed the simulation studies. Z.Z.T. and L.L. wrote the first version of the manuscript. J.S., H.Z., A.C., and D.V.M. also contributed to the writing. All authors provided input and revisions for the final manuscript.

## Competing interests

J.S., H.Z., A.C., and D.V.M. are employees at Merck Sharp & Dohme Corp., a subsidiary of Merck & Co., Inc., Kenilworth, NJ, USA. The remaining authors declare no competing interests.
