## [Peer Review File · Nature Communications]

Reviewers' comments:

Reviewer #1 (Remarks to the Author):

Nice manuscript presenting novel approaches to perform multivariate variant-set tests. The approaches only require summary statistics, both score tests and linkage disequilibrium (LD) matrices, and can be applied to a variety of genetic traits. The application to lipid traits in GLGC is interesting, and the analysis performed is described in sufficient detail to allow someone to reproduce the results. A carefully planned simulation study was performed, evaluating type-1 error and power.

The authors should address the following points:

- 1) The authors comment that the strength of their approach is that no individual-data are required. Authors should compare their approach to MSKAT on individual data. Is there much power loss by not having access to individual level data?
- 2) On page 3, the author states that "Although a handful of methods are available for multi-trait multi-variants tests, they are either designed for common variants or require individual-level genotype and phenotype data." This is not entirely accurate. Please see approaches recently proposed in Chung et al (Chung J, Jun GR, Dupuis J, Farrer LA. Comparison of methods for multivariate gene-based association tests for complex diseases using common variants. *Eur J Hum Genet.* 2019 May;27(5):811-823.). Some of these approaches only require summary statistics. Author should compare their approach to previously proposed methods for multi-trait SNV-based tests.
- 3) On page 5, please clarify which LD measure is used to define the matrix R. There are many measures of LD and authors should clarify which measure is used to define the matrix R.
- 4) On page 7, it is stated that "The matrix B is not estimable from the data and needs to be pre-specified." This is confusing, because the authors propose to use estimates of the genetic correlation from their data in order to pre-specify the B_2 used to define B. Are the genetic correlation estimates from a different study or are they estimated from the data? If estimated from the data, the authors should clarify what they mean by "is not estimable from the data."
- 5) In the application (Supplementary Table 1), the genomic control lambdas are all elevated. Authors should comment on this inflation.
- 6) On page 10, the authors indicate that they compare their tests to minP. How was minP corrected to account for the fact that you are using the minimum P-value over multiple tests to avoid inflated type-I error?
- 7) In Supplementary Table 4, iMSTAT appears to have slightly elevated type-I error. Authors should comment on this and how this affects their conclusion from the power comparison.
- 8) On page 16, authors state that "In addition, future research is required to extend the proposed methods to account for familial and cryptic relatedness." While this appears to be a major limitation of this approach, many software will compute score statistics and variance of the score statistics accounting for cryptic relatedness. Authors should comment on the feasibility of using statistics that already account for familial and cryptic relatedness to compute MSTAR, and how the extension that the authors mention might differ from simply handling in the score statistics and variance computation.

Minor comments:

- 1) Figure S9: Y-axis should say "with finer grid" and not "with more grids".
- 2) Figure S10: Authors should provide estimates for a typical genome-wide analysis in addition to timing study involving a single gene-based test with multi-traits.
- 3) Supplementary table 5: the last 3 columns (Odds.Ratio, Odds.Ratio.CI.Lower, Odds.Ratio.CI.Upper) are all NAs except for one entry. Better to provide information in a footnote for that one database.

Reviewer #2 (Remarks to the Author):

The authors developed MTAR, rare variant association tests for multiple phenotypes using summary statistics. The proposed method provides a valuable tool for boosting the power of RV association tests. It extends the existing commonly used single-phenotype RV association tests, such as SKAT and burden, by leveraging information across multiple phenotypes. MTAR accounts for the correlation among multiple phenotypes by accounting for between-phenotype correlation using their covariance matrix and genetic correlations. Different tests are combined using the Cauchy method to gain robustness. The proposed methods were applied to the lipids exome chip summary statistics in the discovery phase and the UK biobank summary statistics in the replication phase. The paper is clearly written and provides a valuable tool for analyzing multiple phenotype RV association studies.

- The proposed methods iMTAR and cMTAR are similar to Multi-SKAT (Dutta, et al, 2019). This dampens the novelty of the proposed methods.
- The use of the genetic correlation to model the similarity of genetic variant effects on multiple phenotypes is an interesting idea. As genetic correlation is calculated using markers across the genome and RV association tests are performed locally using a variant set. It would be useful to have more discussions on the value of using genetic correlation to model similarity of genetic effects in a local region.
- The proposed method assumes no covariate, and models the correlation of individual variant score statistics using LD. In practice, covariates, such as ancestry PCs, are often used to control for population structure. It would be useful if the authors can extend the proposed methods to accommodate covariates, and discuss how summary statistics of individual markers can be used to accommodate covariates in multiple-phenotype RV association analysis.
- Score statistics are used in constructing RV association test statistics. The authors claimed on page 5 that $\hat{\beta}$'s follow a normal distribution with mean β and covariance Σ . This does not seem to be true in the presence of covariates or binary traits.
- The authors use the UK Biobank GWAS data in the replication phase. This is not ideal given the paper focuses on rare variants. As the WES data of 50K subjects of UK Biobank are available, it would be useful if the authors use the UK biobank WES data in the replication phase.

Reference:

- Dutta, D., Scott, L., Boehnke, M. and Lee, S., 2019. Multi-SKAT: General framework to test for rare-variant association with multiple phenotypes. *Genetic epidemiology*, 43(1), pp.4-23.

Reviewer #3 (Remarks to the Author):

This paper presents a series of rare variant multi-trait association analysis methods, named under MTAR. Gene-based pleiotropy analysis is an important topic. Authors presented results from GLGC Exome chip data and replicated some of the signals in UK Biobank data. The amount of work presented is impressive; however, the presentation of this paper is very confusing and it lacks some key components as a method paper. First, it is almost impossible to follow the descriptions in the introduction and the results because several methods are presented without much justification. I recommend complete restructuring of the manuscript by presenting motivation, philosophy, justification, and application scenarios of each method, not just plainly describe mathematical

formulations of each method. Second, there have been several methods published on rare variant multi-phenotype analysis in recent years, but there is no comparison to the existing methods. To name a few, GAMuT (Broadaway et al), MARV (Kaakinen et al), DKAT (Zhan et al), and MAAUSS (Lee et al). As the paper is primarily a methodological paper, comparing MTAR to these methods, explaining what are the differences and strengths, and some experimental comparisons would be very informative.

- What is the justification of modeling B as Kronecker product of B1 and B2? This part is the most important in the overall design of proposed MTAR methods, but is not properly addressed.

- Page 3, Please define cMTAR, iMTAR and cctP before using abbreviations for the first time. These terms are suddenly introduced without explaining what they are and why they are proposed. In fact, even in Page 6-7 and in Figure 1, cMTAR and iMTAR are just explained by parameters and not explained what are the underlying genetic / biological models.

- Page 6, "the genetic distances are not likely to be the same between any pair of the traits" is unclear. In fact, 'genetic distance' usually refer to the distance between populations or species, not between traits.

- Page 9, 'MTAR P-values for many of these genes are 100-fold more significant than the single-trait-based P-values...' P-values between different models should not be compared to each other to describe one model is more powerful than the other. In fact there are many other places that authors use P-values as a measure of power.

- Is there any sample overlap between GLGC and UK Biobank? GLGC Exome chip data includes many UK samples, hence this might confound the replication analysis.

We would like to thank three reviewers for their comments, which led to a substantial improvement of the paper. Detailed responses to reviewers are given below. We highlighted the changes related to the comments in the revision.

Response to Reviewer #1

1) *The authors comment that the strength of their approach is that no individual-data are required. Authors should compare their approach to MSKAT on individual data. Is there much power loss by not having access to individual level data?*

Response: Thank you for your suggestion. In the revision, we summarized the features of existing multiple-trait multi-variant methods in Supplementary Table 1. The recently developed method Multi-SKAT [1] is shown to be more powerful than MSKAT. Therefore, we compared Multi-SKAT with MTAR in our revision. The simulation results show that Multi-SKAT is less powerful than MTAR in almost all scenarios. In the revised manuscript, the method comparison results were presented in the Results subsection “Comparison with other multi-trait multi-variant methods”.

Research has shown that the meta-analysis of summary statistics is as efficient as the joint analysis of individual-level data [2,3,4,5]. We conducted a simulation study to demonstrate the equivalent efficiency of the MTAR tests using summary statistics and individual-level data. In the simulation, we considered two studies and used summary statistics from the two studies to perform iMTAR, cMTAR, and MTAR-O tests. As an alternative, we pooled the individual-level data from the two studies and used the pooled data to perform tests. The scatter plots below show the two versions of p-values and highlight that the two approaches generate very similar p-values with Pearson correlation coefficient > 0.96 .

[1] Dutta, Diptavo, et al. Multi-SKAT: General framework to test for rare variant association with multiple phenotypes. *Genetic epidemiology* 43.1 (2019): 4-23.

[2] Tang, Z. Z. and Lin, D. Y. (2015) Meta-analysis for discovering rare-variant associations: statistical methods and software programs. *American Journal of Human Genetics*, 97, 3553.

[3] Lin, D. and Zeng, D. (2010) Meta-analysis of genome-wide association studies: no efficiency gain in using individual participant data. *Genetic Epidemiology*, 34(1), 6066.

[4] Lin, D. Y. and Zeng, D. (2010) On the relative efficiency of using summary statistics versus individual-level data in meta-analysis. *Biometrika*, 97(2), 321-332.

[5] Zeng, D. and Lin, D. Y. (2015) On random-effects meta-analysis. *Biometrika*, 102(2), 281-294.

2) *On page 3, the author states that Although a handful of methods are available for multi-trait multi-variants tests, they are either designed for common variants or require individual-level genotype and phenotype data. This is not entire accurate. Please see approaches recently proposed in Chung et al (Chung J, Jun GR, Dupuis J, Farrer LA. Comparison of methods for multivariate gene-based association tests for complex diseases using common variants. Eur J Hum Genet. 2019 May;27(5):811-823.). Some of these approaches only require summary statistics. Author should compare their approach to previously proposed methods for multi-trait SNV-based tests.*

Response: Thank you for pointing out Chung et al.’s paper. In our revision, we have cited the paper and changed the sentence to “Although several methods are available for multi-trait multi-variant tests, most of them are either designed for common variants or require individual-level genotype and phenotype data.”

Chung et al. mentioned that “TATES+GATES (or called MGAS)” can use summary statistics and properly control type I error. MGAS has been shown to have lower power than a method called MTaSPUsSet [6]. Moreover, MTaSPUsSet is more comparable with MTAR because MTaSPUsSet does not require complete trait values for all subjects and can handle any patterns of sample overlap among traits. Therefore, we have compared MTaSPUsSet with MTAR in the revision. The simulation results show that MTaSPUsSet has lower power than MTAR. In the GLGC data analysis, the number of lipid-associated genes identified by MTaSPUsSet is much smaller than MTAR. Moreover, MTaSPUsSet is much more computationally intensive than MTAR because MTaSPUsSet requires MC simulation to get p-values. Please see the Results subsection “Comparison with other multi-trait multi-variant methods” for details.

[6] Kwak, Il-Youp, and Wei Pan. Gene-and pathway-based association tests for multiple traits with GWAS summary statistics. *Bioinformatics* 33.1 (2016): 64-71.

3) *On page 5, please clarify which LD measure is used to define the matrix R. There are many measures of LD and authors should clarify which measure is used to define the matrix R.*

Response: Thank you for your comment. In our revision, we clarified that we used the Pearson correlation coefficient among the genotypes as our LD measure.

4) *On page 7, it is stated that “The matrix \mathbf{B} is not estimable from the data and needs to be pre-specified.” This is confusing, because the authors propose to use estimates of the genetic correlation from their data in order to pre-specify the \mathbf{B}_2 used to define \mathbf{B} . Are the genetic correlation estimates from a different study or are they estimated from the data? If estimated from the data,*

the authors should clarify what they mean by “is not estimable from the data.”

Response: Thank you for your comment. The matrix $\mathbf{B} = \mathbf{B}_2 \otimes \mathbf{B}_1$ is the covariance matrix in the prior distribution we placed on the rare variant effects (please see the hierarchical model presented in Methods on page 22). If our prior reflected the underlying genetic architecture, then the power of the association test would increase. However, the genetic architecture of complex traits is unknown in advance and is likely to vary from one gene to another across the genome and from one trait to another.

The genetic correlation is a “global” measure of the similarity of the genetic effects between traits, which is calculated using the effect estimates of common variants across the genome. In contrast, \mathbf{B}_2 controls the gene-level (“local”) rare-variant effect similarity. Hence, genetic correlation is not the estimate of \mathbf{B}_2 . However, we believe the genetic correlation is (at least partially) informative to reflect the similarity of the gene-level rare-variant associations among traits for most of the genes in the genome and can help to improve the power of multi-trait RVAS. Therefore, in MTAR, \mathbf{B}_2 is specified as some function of genetic correlation with a parameter (ρ_2) to determine the degree of dependency on genetic correlation. In revision, we have rewritten this paragraph to clarify this point.

As pointed out by another reviewer, using genetic correlation to inform \mathbf{B}_2 is one main innovation of MTAR and worth further emphasis. To this end, we modified many places in the paper. We also clarified in the revised paper that the genetic correlation can be conveniently estimated using GWAS summary statistics (in the same study or a different study) and there are web portals for people to query genetic correlations among many complex traits and diseases.

5) In the application (Supplementary Figure 1), the genomic control lambdas are all elevated. Authors should comment on this inflation.

Response: Thank you for your comment. We reanalyzed the GLGC data presented in the paper “Liu, Dajiang J., et al. Exome-wide association study of plasma lipids in > 300,000 individuals. Nature genetics 49.12 (2017): 1758–1766.”. The NG paper focused on sing-variant analysis and did not provide the gene-based test p-values for all the genes and QQ plots. We contacted the first author of the paper and he sent us their gene-based analysis p-values. Below are the QQ plots generated based on their results. As you can see, they have a similar level of genomic control lambdas as ours.

In general, the main reasons that can cause inflated QQ plots are (1) quality control is not properly conducted; (2) population stratification is fully corrected; (3) polygenic inheritance of the trait. We believed that the NG paper has already carefully processed the data to make sure there is no quality control and population stratification issues and the most likely reason is the polygenic inheritance of the lipid traits, which has been demonstrated by many previous publications [e.g. 7, 8]. On page 10, we have commented “Similar to the previous gene-based RVAS of GLGC data, the slightly elevated genomic control lambdas in the quantile-quantile (QQ) plots suggest the polygenic inheritance of the lipid traits”.

[7] Kathiresan, Sekar, et al. Common variants at 30 loci contribute to polygenic dyslipidemia. *Nature genetics* 41.1 (2009): 56.
 [8] Dron, Jacqueline S., and Robert A. Hegele. Genetics of lipid and lipoprotein disorders and traits. *Current genetic medicine reports* 4.3 (2016): 130-141.

6) *On page 10, the authors indicate that they compare their tests to minP. How was minP corrected to account for the fact that you are using the minimum P-value over multiple tests to avoid inflated type-I error?*

Response: Thank you for your comment. On page 10, we clarified that we used the “the Bonferroni-corrected minimal P-value (minP, take the minimal P-values and then multiply it by the number of tests combined)”. Our simulations show that the minP properly controls the type I error (please see Supplementary Table 5).

7) *In Supplementary Table 4, iMTAR appears to have slightly elevated type-I error. Authors should comment on this and how this affects their conclusion from the power comparison?*

Response: Thank you for your comment. We benchmarked our type I error rates against the ACAT paper [9] that also used the Cauchy p-value combination methods for rare-variant association tests. In their Table 1, the maximum inflation in empirical type I error is 1.1 times the α level (e.g. 1.1×10^{-6} at $\alpha = 1.0 \times 10^{-6}$) which is similar to what we have (e.g. 2.74×10^{-6} at $\alpha = 2.5 \times 10^{-6}$). The authors commented that the accuracy of the Cauchy combined p-value is generally satisfactory for practical use, but a slight inflation is possible. We have added this comment to our revision on page 16.

[9] Liu, Yaowu, et al. Acat: A fast and powerful p value combination method for rare-variant analysis in sequencing studies. *The American Journal of Human Genetics* 104.3 (2019): 410-421

8) On page 16, authors state that “In addition, future research is required to extend the proposed methods to account for familial and cryptic relatedness.” While this appears to be a major limitation of this approach, many software will compute score statistics and variance of the score statistics accounting for cryptic relatedness. Authors should comment on the feasibility of using statistics that already account for familial and cryptic relatedness to compute MTAR, and how the extension that the authors mention might differ from simply handling in the score statistics and variance computation.

Response: Thank you for your comment. Indeed, we can use the currently available methods and software programs to generate valid score statistics and their covariance matrices for family data and use them in MTAR. The main difficulty lies in handling sample overlap across traits. As you can see in Methods (on page 21-22), the derivation of the formula for computing $\text{cov}(\widehat{\beta}_k, \widehat{\beta}_{k'})$ starts from the models for independent data. It is not clear if the formula still holds under the models for related observations. We think this needs further investigation. In our revision, we have clarified this in our description of the future research on page 19.

Minor comments:

- 1) *Figure S9: Y-axis should say with finer grid and not with more grids.*
- 2) *Figure S10: Authors should provide estimates for a typical genome-wide analysis in addition to timing study involving a single gene-based test with multi-traits.*
- 3) *Supplementary table 5: the last 3 columns (Odds.Ratio, Odds.Ratio.CI.Lower, Odds.Ratio.CI.Upper) are all NAs except for one entry. Better to provide information in a footnote for that one database.*

Response: Thank you. We have incorporated these changes into the revision. For 2), the estimate of time for a genome-wide analysis has been provided the highlighted text on page 26.

Response to Reviewer #2

1) *The proposed methods iMTAR and cMTAR are similar to Multi-SKAT (Dutta, et al, 2019). This dampens the novelty of the proposed methods.*

Response: Thank you for pointing out the Multi-SKAT paper. In the revision, we have summarized the features of existing multiple-trait multi-variant methods in Supplementary Table 1. As you can see, Multi-SKAT requires individual-level genotype and phenotype data and cannot use summary statistics. This is a big limitation since most genetics consortia only share summary statistics but not the individual-level data due to privacy concerns and the size of the whole genome/exome datasets. Therefore, the meta-analysis of summary statistics is strongly preferable to the analysis of individual-level data.

In addition, Multi-SKAT requires complete trait values for all the subjects and cannot handle complex patterns of sample overlap across traits (e.g. in GLGC, each trait is only measured in a subset of cohorts but not all the cohorts).

In our revision, we compared Multi-SKAT with our proposed method MTAR in simulation studies. The simulation results show that Multi-SKAT is less powerful than MTAR in almost all scenarios. The method comparison results were presented in the Results subsection “Comparison with other multi-trait multi-variant methods”.

2) *The use of the genetic correlation to model the similarity of genetic variant effects on multiple phenotypes is an interesting idea. As genetic correlation is calculated using markers across the genome and RV association tests are performed locally using a variant set. It would be useful to have more discussions on the value of using genetic correlation to model similarity of genetic effects in a local region.*

Response: Indeed, MTAR gains a substantial power by leveraging the genome-wide genetic correlation measure to inform the degree of gene-level RV effect heterogeneity across traits. This is one main innovation of the method that no existing methods have considered. We have modified many places throughout the paper to emphasize this important point. In particular, we performed an additional test in the GLGC analysis simply using the exchangeable correlation structure in \mathbf{B}_2 . About 18% of the discovered genes would become insignificant in this test. This fact strongly demonstrates the effectiveness of this strategy in gaining power.

3) *The proposed method assumes no covariate, and models the correlation of individual variant score statistics using LD. In practice, covariates, such as ancestry PCs, are often used to control for population structure. It would be useful if the authors can extend the proposed methods to accommodate covariates, and discuss how summary statistics of individual markers can be used to accommodate covariates in multiple-phenotype RV association analysis.*

Response: We agree that it is important for a method to have the capability to accommodate covariates. The proposed methods have already considered covariates. As shown on page 20, the score statistics we have are derived from the model that has adjusted covariates in \mathbf{X}_{ik} .

The use of single variant score statistics and LD matrix to recover gene-level score statistics has been justified in previous research (e.g. please see [1], we have cited this paper). In their proof, covariates are allowed in the model. In our simulation studies, we included two covariates (X_{i1} is binary and X_{i2} is continuous) in our model to simulate traits (please see page 26).

[1] Hu, Yi-Juan, et al. Meta-analysis of gene-level associations for rare variants based on single-variant statistics. The American Journal of Human Genetics 93.2 (2013): 236-248.

4) *Score statistics are used in constructing RV association test statistics. The authors claimed on page 5 that beta-hats follow a normal distribution with mean beta and covariance Sigma. This does not seem to be true in the presence of covariates or binary traits.*

Response: The normality of score statistics or beta-hats is still true in the presence of covariates and binary traits. The majority of rare-variant association tests use the score statistics and the normality of the score statistics is the basis for all these rare variants association tests [2]. In addition, previous research [3,4] has shown that the $\hat{\beta}_k$ defined as the rescaled score statistics (i.e. $\hat{\beta}_k = \mathbf{V}_k^{-1}\mathbf{U}_k$) also follows the normal distribution under generalized linear model (GLM, therefore including the case of a binary trait) in the presence of covariates. In the revision, we have cited these papers the first time we mentioned $\hat{\beta}_k$ (on page 6).

[2] Lee, Seunggeung, et al. Rare-variant association analysis: study designs and statistical tests. The American Journal of Human Genetics 95.1 (2014): 5-23.

[3] Tang, ZhengZheng, and DanYu Lin. Metaanalysis of sequencing studies with heterogeneous genetic associations. Genetic epidemiology 38.5 (2014): 389-401.

[4] Tang, Zheng-Zheng, and Dan-Yu Lin. Meta-analysis for discovering rare-variant associations: statistical methods and software programs. The American Journal of Human Genetics 97.1 (2015): 35-53.

5) *The authors use the UK Biobank GWAS data in the replication phase. This is not ideal given the paper focuses on rare variants. As the WES data of 50K subjects of UK Biobank are available, it would be useful if the authors use the UK biobank WES data in the replication phase.*

Response: The UK Biobank WES data of 50K subjects have been released, but errors have been identified within the 50k FE exome data release. UK Biobank's position is that rather than try to re-issue the 50k FE dataset, it would be more beneficial for researchers to have UK Biobank concentrate its efforts on the release of the first 150k exomes. This release is currently scheduled for spring 2020 and will include the re-release of the corrected 50k exome datasets. The schedule is simply too late for our timeline.

Although we used UK Biobank GWAS data, we still focused on the rare variants (MAF < 5%). We also tried a more conservative approach in the replication: only include rare variants that are present in both UK Biobank and GLGC. Therefore, the number of variants in the replication stage is generally smaller than the discovery stage and this subset of the variants may miss some causal variants in a given gene. Nevertheless, we still successfully replicated 7 out of 11 novel genes.

Response to Reviewer #3

1) *I recommend complete restructuring of the manuscript by presenting motivation, philosophy, justification, and application scenarios of each method, not just plainly describe mathematical formulations of each method.*

Response: Thank you for your valuable comments on the writing. We have revised the manuscript according to your suggestions and highlighted changes. We think the revision clearly delivers the ideas of the methods and greatly improves the presentation.

2) *There have been several methods published on rare variant multi-phenotype analysis in recent years, but there is no comparison to the existing methods. To name a few, GAMuT (Broadaway et al), MARV (Kaakinen et al), DKAT (Zhan et al), and MAAUSS (Lee et al). As the paper is primarily a methodological paper, comparing MTAR to these methods, explaining what are the differences and strengths, and some experimental comparisons would be very informative.*

Response: Thank you for pointing out these papers. In the revision, we have summarized the features of existing multiple-trait multi-variant methods in Supplementary Table 1. As you can see from the table, GAMuT, MARV, DKAT, and MAAUSS belong to a category of the methods that require individual-level genotype and phenotype data and cannot take summary statistics in the analysis. This is a big limitation since most genetics consortia only share summary statistics but not the individual-level data due to privacy concerns and the size of the whole genome/exome datasets. Therefore, the meta-analysis of summary statistics is strongly preferable to the analysis of individual-level data.

In addition, these methods require complete trait values for all the subjects and cannot handle complex patterns of sample overlap across traits (e.g. in GLGC, each trait is only measured in a subset of cohorts but not all the cohorts).

In our revision, we compared Multi-SKAT [1] with our proposed method MTAR in simulation studies because Multi-SKAT is the most recently developed method in the category of methods that requires individual-level data and had been shown to have higher power than many other methods in the same category (e.g. GAMuT, DKAT, and MAAUSS). The simulation results show that Multi-SKAT is less powerful than MTAR in almost all simulation settings.

Besides Multi-SKAT, we considered another method called MTaSPUsSet [2]. This method is more comparable with MTAR because it can use summary statistics and handle any patterns of sample overlap among traits. The simulation results show that MTaSPUsSet is less powerful than MTAR methods in all simulation settings. In the GLGC data analysis, the number of lipid-associated genes identified by MTaSPUsSet is much smaller than MTAR. Moreover, MTaSPUsSet is much more computationally intensive than MTAR because MTaSPUsSet requires MC simulation to obtain p-values.

All comparison results were presented in the Results subsection Comparison with other multi-trait multi-variant methods.

[1] Dutta, Diptavo, et al. "MultiSKAT: General framework to test for rarevariant association with multiple phenotypes." Genetic epidemiology 43.1 (2019): 4-23.

[2] Kwak, Il-Youp, and Wei Pan. "Gene-and pathway-based association tests for multiple traits with GWAS summary statistics." Bioinformatics 33.1 (2016): 64-71.

3) *What is the justification of modeling \mathbf{B} as Kronecker product of \mathbf{B}_1 and \mathbf{B}_2 ? This part is the most important in the overall design of proposed MTAR methods, but is not properly addressed.*

Response: Thank you for your comment. The matrix \mathbf{B} is the covariance matrix in the prior distribution we placed on the rare variant effects (please see the hierarchical model presented in Methods on page 22). Therefore, the misspecification of \mathbf{B} will not affect the validity of the resulting association tests.

If the specification of \mathbf{B} reflected the underlying genetic architecture, then the power of the association test would increase. However, the genetic architecture of complex traits is unknown in advance and is likely to vary from one gene to another across the genome and from one trait to another. Therefore, the main challenge of multi-trait multi-variant tests is to flexibly accommodate a variety of genetic effect patterns among traits and variants such that the test is robust and has high power. Factoring the full covariance \mathbf{B} as $\mathbf{B}_2 \otimes \mathbf{B}_1$ simply enables us to separately model the genetic structures among traits and among variants. In MTAR, we use different correlation structures in \mathbf{B}_1 and \mathbf{B}_2 to represent a wide spectrum of association patterns across traits and variants.

In our revised manuscript (Introduction and Methods subsection "MTAR overview"), we have clearly explained how we specify \mathbf{B}_1 and \mathbf{B}_2 and their corresponding genetic models.

4) *Page 3, Please define $cMTAR$, $iMTAR$ and $cctP$ before using abbreviations for the first time. These terms are suddenly introduced without explaining what they are and why they are proposed. In fact, even in Page 6-7 and in Figure 1, $cMTAR$ and $iMTAR$ are just explained by parameters and not explained what are the underlying genetic / biological models.*

Response: Thank you for your suggestion. In the revision, we have defined the tests before using abbreviations for the first time and discussed why they are proposed in the Introduction. In the Methods subsection "MTAR overview", we have explained the underlying genetic models/effect patterns and modified Figure 1. We have highlighted these changes in the revision.

5) *Page 6, "the genetic distances are not likely to be the same between any pair of the traits" is unclear. In fact, 'genetic distance' usually refer to the distance between populations or species, not between traits.*

Response: Thank you for pointing that out. We have changed the sentence to "It is not sensible to assume an exchangeable correlation structure for \mathbf{B}_2 because some pairs of traits

are more similar in the rare variant effects than other pairs (e.g. the two diseases that were caused by the same set of rare mutations would have a larger correlation in their rare variant genetic effects).” We have also explained this point in the Introduction (please see the 2nd paragraph on page 4).

6) Page 9, “MTAR *P*-values for many of these genes are 100-fold more significant than the single-trait-based *P*-values...” *P*-values between different models should not be compared to each other to describe one model is more powerful than the other. In fact there are many other places that authors use *P*-values as a measure of power.

Response: Thank you for your comment. We agree that *p*-values do not “measure” power since we do not know the true genetic model in real data analysis. However, we think *p*-values do “reflect” power given all the methods under comparison properly control the type I error rates under the null model (i.e. no association). For example, in the highly cited paper of the SKAT method [3], the authors compared SKAT with other rare-variant association tests in the analysis of Dallas Heart Study Sequencing Data (Table 4 in the SKAT paper). Commenting *p*-values from various methods in their data analysis, the authors said “*SKAT was by far the most powerful test for the dichotomous trait. For continuous traits, SKAT has much smaller p values than two burden methods (CAST and WST) and VT*”. Similar to the SKAT paper, we think the observation that MTAR usually produces much smaller *p*-value than the single-trait-based tests suggests the good performance of MTAR. To avoid confusion, we have changed “100-fold more significant” to “100-fold smaller than” and rephrased several other sentences in the section of real data analysis.

[3] Wu, Michael C., et al. “Rare-variant association testing for sequencing data with the sequence kernel association test.” *The American Journal of Human Genetics* 89.1 (2011): 82-93.

7) *Is there any sample overlap between GLGC and UK Biobank? GLGC Exome chip data includes many UK samples, hence this might confound the replication analysis.*

Response: In our paper, we reanalyzed the version of GLGC data in “Liu, Dajiang J., et al. Exome-wide association study of plasma lipids in > 300,000 individuals. *Nature genetics* 49.12 (2017): 1758–1766.”. In the NG paper, UK Biobank was not one of the GLGC cohorts (please see a full list of GLGC cohorts in the Supplementary File of that paper). In addition, we contacted the first author of the paper and he confirmed that UK Biobank is not part of GLGC cohort for calculating the summary statistics we used on the web portal <http://csg.sph.umich.edu/abecasis/public/lipids2017/>. In fact, similar to our paper, the NG paper used UK Biobank data to replicate the associations for some genes discovered from the analysis of GLGC. We hope this clarifies the validity of the replication analysis.

REVIEWERS' COMMENTS:

Reviewer #1 (Remarks to the Author):

The authors have extensively revised the manuscript to satisfactorily address all reviewers's comments. They have added extensive comparisons to other approaches, and have justified to use of UK Biobank imputed data, resulting in a much improved manuscript. I have no further comments or suggestions.

Reviewer #3 (Remarks to the Author):

The authors addressed most of my concerns by adding detailed description on the methods and making comparisons to prior methods. I do not have further comments.

Response to Reviewers

We would like to thank the reviewers for their time to evaluate our revised manuscript. We are glad that the reviewers find our revisions satisfactory and our manuscript much improved.

REVIEWERS' COMMENTS:

Reviewer #1:

The authors have extensively revised the manuscript to satisfactorily address all reviewers's comments. They have added extensive comparisons to other approaches, and have justified to use of UK Biobank imputed data, resulting in a much improved manuscript. I have no further comments or suggestions.

Reviewer #2:

The authors addressed most of my concerns by adding detailed description on the methods and making comparisons to prior methods. I do not have further comments.

Sincerely,

Zheng-Zheng Tang
Assistant Professor of Biostatistics and Medical Informatics
University of Wisconsin-Madison